# The crystal structure of Cry78Aa from *Bacillus thuringiensis* provides insights into its insecticidal activity

Beibei Cao [1], Yangfan Nie[2], Zeyuan Guan [2], Chuanyu Chen[2], Nancong Wang[2], Zeyu Wang[1], Changlong Shu [1], Jie Zhang [1✉] & Delin Zhang [2✉]

Genetically modified plants with insecticidal proteins from *Bacillus thuringiensis* (Bt) have been successfully utilized to control various kinds of pests in crop production and reduce the abuse of pesticides. However, a limited number of genes are available for the protection of crops from rice planthopper. Recently, Cry78Aa protein from Bt strain C9F1 has been found to have high insecticidal activity against *Laodelphax striatellus* and *Nilaparvata lugens*. It is the first reported single-component protein in the world to combat rice planthoppers, making it very promising for use in transgenic crops. The ambiguous mechanism of Cry78Aa functions prevented further engineering or application. Here, we report the crystal structure of Cry78Aa, which consists of two domains: a C-terminal β-pore forming domain belonging to the aerolysin family and an N-terminal trefoil domain resembling the S-type ricin B lectin. Thus, Cry78Aa could represent a distinctive type of β-pore forming toxin. We also found that Cry78Aa binds carbohydrates such as galactose derivatives and is essential for insecticidal activity against *Laodelphax striatellus*. Our results suggest a mechanism underlying the function of Cry78Aa against rice planthoppers and pave the way to maximizing the usage of the toxin.

[1] State Key Laboratory for Biology of Plant Diseases and Insect Pests, Institute of Plant Protection, Chinese Academy of Agricultural Sciences, Beijing 100193, China. [2] National Key Laboratory of Crop Genetic Improvement, Hubei Hongshan Laboratory, Huazhong Agricultural University, Wuhan 430070, China. ✉email: zhangjie05@caas.cn; zdl@mail.hzau.edu.cn

Rice is one of the most important staple food crops worldwide, especially in Asia. Rice planthoppers are a major pest that causes significant yield loss, not only by sucking phloem sap but also spreading viruses[1]. Currently, the control of rice planthoppers in crop production relies mainly on the application of chemical pesticides, but the unscientific use of chemical pesticides can lead to increased resistance of the pest against pesticides and cause damage to beneficial insects and the environment[2,3]. Biological control methods using *Bacillus thuringensis* (Bt) as well as genetically modified plants expressing insecticidal proteins from Bt have been proven effective and economic against some insect pests[4]. The main genetically modified crops planted worldwide are corn, cotton, soybeans, and rapeseed[5]. In the future, transgenic Bt rice is expected to be officially used for commercial planting to control rice planthoppers and other pests. However, the development of biological control against planthoppers remains limited due to the deficiency of gene sources. Thus far, only Cry64Ba/Cry64Ca[6] and modified Cry1Ab[7] have been reported to possess considerable toxicity against rice planthoppers. The discovery of new toxins with higher efficiency targeting rice planthoppers will hopefully accelerate the development of biological control of this economically important pest.

Cry78Aa is a novel protein identified from the Bt C9F1 strain that effectively kills rice planthoppers, with median lethal concentration ($LC_{50}$) values against *Laodelphax striatellus* and *Nilaparvata lugens* of 6.89 and 15.78 μg ml$^{-1}$, respectively[8]. The activity of Cry78Aa does not require in vitro activation or any additional components, making it convenient for application in field trials. The current research on the crystal structure of Cry proteins revolves mainly around the three-domain Cry (3D-Cry) protein with high insecticidal activity against Lepidoptera and Coleoptera[9], while structures of insecticidal toxins that are active against Hemiptera pests such as rice planthoppers are rarely reported. Only the Cry51Aa1 protein (PDB: 4PKM) and SeMet-Cry51Aa2-L11M (PDB: 5HD2) which have insecticidal activity against *lygus*, were analyzed[10,11]. In addition, some aerolysin-type β-pore-forming toxins belonging to the Bt family have been reported, including Cry35Ab1 (PDB: 4JP0), which combines with Cry34Ab1 and has activity against western corn rootworm[12], the mosquito toxin BinAB (PDB: 5FOY)[13], and parasporin-2 (PDB: 2ZTB), which is toxic to cancer cells[14].

Sequence analysis of Cry78Aa suggests that it has an architecture similar to that of the Bin-like toxins, which contain an N-terminal β-trefoil domain predicted to be involved in receptor binding and a C-terminal Toxin_10 domain believed to be the pore-forming domain[8]. Structural prediction of Cry78Aa based on the resolved structure of a homologous Cry35Ab1 toxin implies that it fits the 'head and tail' model of other β-pore-forming toxins with pesticidal activity[12]. Nevertheless, solid structural information on Cry78Aa is required not only to understand the mechanism underpinning the insecticidal activity of Cry78Aa but also to explain, to a certain degree, the low activity of 3D-Cry protein against rice planthoppers and other sucking pests. More meaningfully, we can design and develop engineered mutants with higher activity against sucking-type pests (such as rice planthoppers, aphids, stink bugs) based on the structural information of Cry78Aa, which can be further applied to the safe production of crops.

Here, we resolved the crystal structure of Cry78Aa$_{13-359}$, which approximately resembles full-length Cry78Aa. This structure consists of two independent domains: a trefoil domain at the N-terminus, which shares the highest identity with S-type lectin, and a pore-forming domain belonging to the aerolysin family. We also resolved the structure of Cry78Aa$_{155-359}$, which resembles the solitary pore-forming domain, and found that it has an identical conformation with Cry78Aa$_{13-359}$. Bioassays showed that the NTD or CTD of Cry78Aa alone has no toxicity against planthopper nymphs, indicating that its insecticidal activity is dependent on the cooperation of both domains. The NTD of Cry78Aa plays a vital role for its insecticidal activity, probably by recognize galactose derivatives linked to proteins or lipids on the surface of the cell membrane. A model was proposed to infer the process by which Cry78Aa functions based on structural analysis and biochemical results.

## Results

**The primary structure of the Cry78Aa protein is obviously different from that of the 3D-Cry protein**. Phylogenetic tree analysis divided 311 model Cry proteins into seven main clades (Supplementary Fig. 1). Cry78Aa (red label) is clustered as Clade 2 (golden branches) containing all insecticidal proteins homologous to the Toxin_10/Bin family (Tpp). In the Tpp family, only the three-dimensional structure of the Cry35Ab1 protein that forms a binary toxin with Cry34Ab1 has been reported. In addition, all 3D-Cry proteins are clustered as Clade 7 (black branches). Cry78Aa is obviously different from Clade 7 of proteins, and the relationship is farther. Therefore, we sought to resolve the structure of Cry78Aa to determine whether it adopts a novel structure compared to other Bt toxin proteins.

**Structural determination of Cry78Aa$_{13-359}$ and Cry78Aa$_{155-359}$**. We used a Cry78Aa protein from Bt strain C9F1, named Cry78Aa1 (GenBank accession No. KY780623), in this study. Full-length Cry78Aa1 consists of 359 amino acids, as detected by mass spectrometry in a spore/crystal mixture of strain C9F1[8]. It is not clear whether the toxin needs to be digested by insect midgut juice to adopt an active form. We initiated our crystallization screening using N-terminal or C-terminal His-tagged full-length Cry78Aa, as well as untagged Cry78Aa prepared by digestion of protease drICE. Only the untagged Cry78Aa yielded different shapes of crystals under 37 conditions. After numerous crystal optimization trials with various precipitants, buffers, salts, and additives during the crystallization procedure, the best crystals diffracted to only 3.5 Å, which is insufficient for structure determination, and could not be further improved. By applying limited proteolysis with several commercial proteases (Supplementary Fig. 2), a stable band slightly smaller than the full-length Cry78Aa band could be observed. We proposed that this stable band was missing ~10 amino acids from the N-terminal or C-terminal of the full-length Cry78Aa and would be a good substitute for crystal obtainment and structure determination. After screening and optimizing the crystalization conditions of several truncated Cry78Aa proteins, the construct Cry78Aa$_{13-359}$ yielded a diamond-shaped crystal with good diffraction data. The data could not be resolved into structures using the coordinates available in the PDB database through the molecular replacement method. We then employed single-wavelength anomalous diffraction (SAD) by growing a derivative crystal of Cry78Aa$_{13-359}$ with selenomethionine. However, only the N-terminal part of Cry78Aa$_{13-359}$ could be determined because its crystal has a very large unit cell.

To obtain the structure of this nearly full-length Cry78Aa, the coordinates of its C-terminal region needed to be resolved first. Using crystallization screening of a few constructs, we identified one condition that yielded a small cube-shaped crystal of Cry78Aa$_{155-359}$. After growing and optimizing the selenomethionine derivative crystal of Cry78Aa$_{155-359}$, its structure was determined by the SAD method and refined to 2.4 Å resolution with $R_{work}$ and $R_{free}$ values of 0.198 and 0.249, respectively (Table 1). Consequently, the structure of Cry78Aa$_{13-359}$ was

**Table 1 Statistics of data collection and refinement.**

| Structure | Cry78Aa$_{13-359}$ (7Y78) | Cry78Aa$_{155-359}$ (7Y79) |
|---|---|---|
| Space group | P 2 21 21 | C2 |
| Unit cell (Å,°) | 66.94 132.74 314.68 | 65.96 76.00 89.03 |
|  | 90.00 90.00 90.00 | 90.00 111.42 90.00 |
| Wavelength (Å) | 0.9789 | 0.9785 |
| Resolution (Å) | 45~2.90 | 45~2.32 |
|  | (2.97-2.90) | (2.40-2.32) |
| $R_{merge}$ (%) | 10.2 (70.2) | 8.8 (72.2) |
| $R_{pim}$ (%) | 4.3 (29.2) | 3.7 (29.8) |
| $I/\sigma(I)$ | 12.6 (2.7) | 13.6 (2.7) |
| Completeness (%) | 99.7 (93.3) | 99.6 (97.2) |
| Number of measured reflections | 415,436 (28,593) | 118,326 (11,549) |
| Number of unique reflections | 63,102 (4303) | 17,763 (1724) |
| Redundancy | 6.6 (6.6) | 6.7 (6.7) |
| Wilson B factor (Å$^2$) | 54.4 | 41.1 |
| $R_{work}$ /$R_{free}$ (%) | 27.92/28.22 | 19.82/24.92 |
| Number of atoms |  |  |
| Protein main chain | 5440 | 1532 |
| Protein side chain | 5453 | 1571 |
| Protein all atoms | 10,893 | 3103 |
| Water molecules | 0 | 85 |
| Other entities | 20 | 0 |
| All atoms | 10,913 | 3188 |
| Average B value (Å$^2$) |  |  |
| Protein main chain | 80.6 | 49.1 |
| Protein side chain | 83.8 | 52.6 |
| Protein all atoms | 82.2 | 50.9 |
| Water molecules | 0 | 47.5 |
| Other entities | 71.4 | 0 |
| All atoms | 82.2 | 50.8 |
| Rms deviations from ideal values |  |  |
| Bonds (Å) | 0.008 | 0.008 |
| Angle (°) | 0.727 | 0.848 |
| Ramachandran plot statistics (%) |  |  |
| Most favorable | 98.82 | 97.36 |
| Additionally allowed | 1.18 | 2.64 |
| Generously allowed | 0 | 0 |
| Disallowed | 0 | 0 |

Values in parentheses are for the highest resolution shell. $R_{merge} = \Sigma_h\Sigma_i|I_{h,i} - I_h|/\Sigma_h\Sigma_i I_{h,i}$, where $I_h$ is the mean intensity of the $i$ observations of symmetry-related reflections of $h$. $R = \Sigma|F_{obs} - F_{calc}|/\Sigma F_{obs}$, where $F_{calc}$ is the calculated protein structure factor from the atomic model ($R_{free}$ was calculated with 5% of the reflections selected).

of 31.3, and the second was chain A of Cry35Ab1 (PDB: 4JP0) with a Z score of 29.8. Both BinB and Cry35Ab1 are composed of an N-terminal trefoil domain and a C-terminal β pore-forming domain, which are typical characteristics of aerolysin family toxins[12,16]. We subsequently aligned the structure of Cry78Aa$_{13-359}$ with BinB and Cry35Ab1 (Supplementary Fig. 3d, e). It appeared that both BinB and Cry35Ab1 have an overall identical conformation to that of Cry78Aa$_{13-359}$. However, the C-terminal domain (CTD) of Cry78Aa shares greater similarity with those of BinB and Cry35Ab1 compared with the N-terminal domains (NTDs), in terms of both structure and sequence identity (Supplementary Fig. 3d–f).

**The domain organization and activity of Cry78Aa**. Taking one protomer (Mol A) as an example, monomeric Cry78Aa$_{13-359}$ exhibits a rigid rod-like shape and comprises two domains, the NTD and the CTD. The NTD of Cry78Aa$_{13-359}$ glues four protein molecules together. The NTD of Cry78Aa$_{13-359}$ has a typical trefoil shape which is usually composed of three sets of β-sheets and is also known as the trefoil domain (Fig. 2a). The CTD of Cry78Aa$_{13-359}$ is mainly composed of several long parallel β-sheets and three α-helices, and is regarded as a β-pore-forming domain. The structure of Cry78Aa$_{155-359}$ represents the conformation of the CTD in the absence of the NTD and has a domain organization similar to that in Cry78Aa$_{13-359}$ (Fig. 2b). According to the structure of BinA/B, which has significant structural similarity with Cry78Aa, the CTD of Cry78Aa can be further divided into the sheet domain, TM domain and sandwich domain (Fig. 2a, b). Structural comparison of Cry78Aa$_{13-359}$ and Cry78Aa$_{155-359}$ indicated that the β-pore-forming domain in the presence or absence of the NTD was nearly superimposed (Supplementary Fig. 3c), suggesting that the absence of the NTD of Cry78Aa did not result in a dramatic conformational change in the CTD pore-forming domain.

To investigate whether the NTD or CTD of Cry78Aa alone is sufficient to exert its insecticidal activity against *L. striatellus* nymphs, we employed a bioassay to assess their function. As expected, full-length Cry78Aa has significant insecticidal activity against *L. striatellus* (Fig. 2c; Supplementary Data 1), as the addition of only 25 μg ml$^{-1}$ protein in forage can result in the death of almost all third-instar nymphs. The construct Cry78Aa$_{1-151}$, which represents the NTD of Cry78Aa alone, has no detectable activity toward the nymphs, as the addition of 25 μg ml$^{-1}$ Cry78Aa$_{1-151}$ into forage did not show any effect on the survival of the nymphs (Fig. 2c; Supplementary Data 1). Similar results were observed when testing the effects of the pore-forming domain Cry78Aa$_{155-359}$ on the survival of the nymphs: nearly all the nymphs in the assay remained alive under the same conditions (Fig. 2c; Supplementary Data 1). These results indicate that neither the trefoil domain nor the pore-forming domain of Cry78Aa alone can trigger the death of *L. striatellus*.

The NTD and CTD of Cry78Aa are covalently connected by a long loop extending from Q162 to N190 as observed in its structure (Fig. 2a; Supplementary Fig. 3f). The trefoil domain of Cry78Aa has a considerable interaction interface with its pore-forming domain. The interface has an area of ~781.2 Å$^2$ and contains 22 residues from the NTD and 20 residues from the CTD (Supplementary Fig. 4a). These residues form an extensive interaction network primarily formed by polar contacts including hydrogen bonds (Supplementary Fig. 4b, c) and van der Waals interactions (Supplementary Fig. 4d, e). This observation implies that the NTD of Cry78Aa has a direct interaction with its CTD. We co-expressed the individual NTD and CTD of Cry78Aa and found that they could be coeluted during affinity chromatography, where the His-tagged CTD of Cry78Aa pulled down its

resolved by combining SAD with molecular replacement using Cry78Aa$_{155-359}$ as a model. The final structure of Cry78Aa$_{13-359}$ was refined to 2.9 Å resolution with $R_{work}$ and $R_{free}$ values of 0.279 and 0.282, respectively (Table 1). Cry78Aa$_{13-359}$ exists as an X-shaped tetramer in the crystal structure and the four molecules are nearly identical to each other (Fig. 1a; Supplementary Fig. 3a). Mol A and Mol B are arranged in one plane, and Mol C and Mol D symmetrically protrude from the plane by ~15 degrees (Fig. 1a). Cry78Aa$_{155-359}$ packed as a dimer in the crystal, and its two protomers were almost superimposed on each other (Fig. 1b; Supplementary Fig. 3b). Structural alignment of Cry78Aa$_{13-359}$ (Mol A) and Cry78Aa$_{155-359}$ (Mol B) revealed nearly identical conformations, with a root-mean-square deviation (RMSD) of 0.566 Å over 175 Cα atoms (Supplementary Fig. 3c).

We ran a Dali search[15] using the coordinates of Cry78Aa$_{13-359}$ and found two structures sharing notably high identity with Cry78Aa$_{13-359}$ (Supplementary Table 1). The first was chain B of the binary mosquito larvicide BinAB (PDB: 5FOY), with a Z score

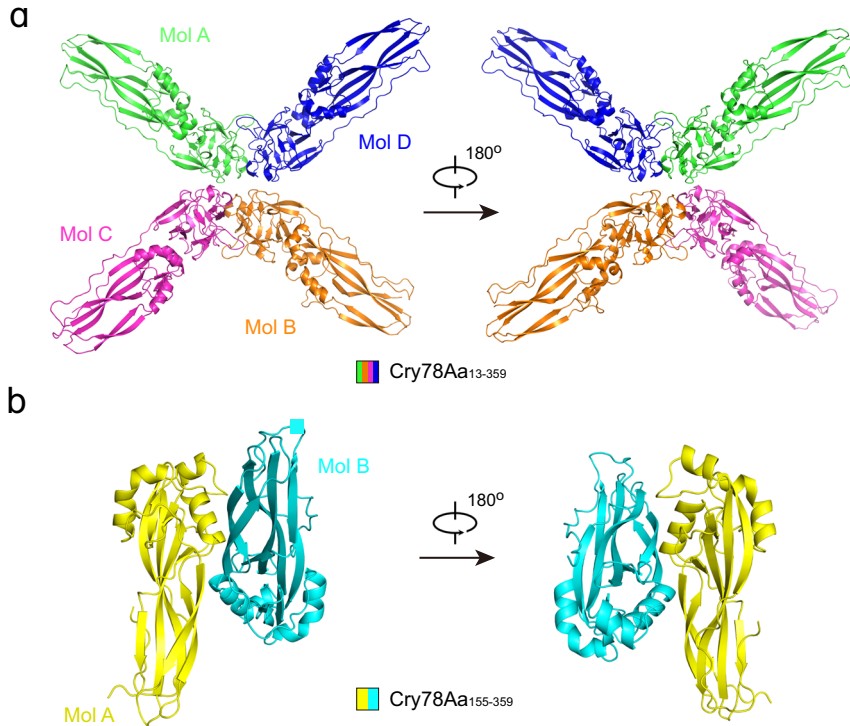

**Fig. 1 The overall structure of Cry78Aa$_{13-359}$ and Cry78Aa$_{155-359}$ in crystals. a** Cartoon representation of Cry78Aa$_{13-359}$, which was packed as an X-shaped tetramer in the crystal. Four protomers are shown as Mol A (green), Mol B (orange), Mol C (magenta), and Mol D (blue). Mol A and Mol B are arranged in a straight line and are embedded in the plane of the page. Mol C and Mol D symmetrically protrude from the plane by ~15°. The view was rotated 180° to provide more information on Cry78Aa$_{13-359}$. **b** Cartoon representation of Cry78Aa$_{13-359}$, which was packed as a dimer in the crystal. Two protomers are shown as Mol A (yellow) and Mol B (cyan). The structure is presented from two symmetrical perspectives.

untagged NTD for purification (Supplementary Fig. 4f). However, when nymphs of *L. striatellus* were fed with both the separately purified NTD and CTD of Cry78Aa, the insecticidal activity was not recovered (Fig. 2c; Supplementary Data 1). This may result from unsuccessful reconstitution of functional Cry78Aa by the NTD and CTD in vitro.

**Oligomerization state of Cry78Aa in solution**. Structural analysis suggested that Cry78Aa$_{13-359}$ and Cry78Aa$_{155-359}$ could exist as oligomers under certain condition, yet it is not clear whether the oligomerization resulted from regular packing of the protein in crystal growth or was an innate property of the protein. During protein purification by gel filtration, Cry78Aa$_{13-359}$ and Cry78Aa$_{155-359}$ eluted mainly as a homogeneous monomer in Tris-HCl pH 8.0 buffer. To further verify the state of Cry78Aa in solution and under native conditions, we employed a static light scattering method to measure their molecular masses. The calculated molecular weight (MW) of the full-length Cry78Aa$_{1-359}$ was ~37.7 kDa (Fig. 2d), which was consistent with the theoretical MW of monomeric Cry78Aa (40.7 kDa). A similar result was observed for Cry78Aa$_{13-359}$. Cry78Aa$_{155-359}$ had an apparent MW of 23.7 kDa (Fig. 2d), which was in accordance with its monomeric state of Cry78Aa$_{155-359}$ (23.7 kDa). These results suggest that both full-length Cry78Aa$_{1-359}$ and Cry78Aa$_{155-359}$ exist mainly as monomers in solution.

Despite the monomeric protein being the main form during gel filtration, Cry78Aa has a propensity to oligomerize in solution. A shoulder with low signal intensity in the gel filtration (SD200) profile of the full-length Cry78Aa, which eluted at ~16 ml volume, could be the oligomeric form of Cry78Aa (Supplementary Fig. 5a). SDS–PAGE and Coomassie brilliant blue staining

results confirmed that the shoulder was indeed Cry78Aa (Supplementary Fig. 5b). The oligomeric and monomeric Cry78Aa probably have distinct shapes in solution because their negative staining appearance detected by transmission electron microscope (TEM) was significantly different. The oligomeric state of Cry78Aa exhibits a fan-like exterior, and the monomeric state resembles a rod, which is consistent with the resolved crystal structure of Cry78Aa in shape and size (Supplementary Fig. 5c, d). Cry78Aa$_{1-151}$, which represents the NTD of Cry78Aa, eluted at ~10 ml volume in SD200 gel filtration, implying that it organized into a high-order structure (Supplementary Fig. 5e, f). In addition, the CTD of Cry78Aa showed a strong tendency to aggregate under mild denaturing conditions. SDS–PAGE detection revealed that monomeric Cry78Aa$_{155-359}$ can easily transform into oligomers or aggregates in 0.1% SDS, with ~30% of the total sample loaded onto each lane (Supplementary Fig. 5g). These results suggested that Cry78Aa is prone to oligomerization under suitable conditions.

**Cry78Aa oligomerized when incubated with BBMVs from *Laodelphax striatellus***. To analyze the oligomerization status of Cry78Aa protein under physiological conditions, the oligomer formation of Cry78Aa was studied by incubating Cry78Aa protein with brush border membrane vesicles (BBMVs) from the sensitive pest *Laodelphax striatellus*. After incubation with BBMVs, obvious oligomers with molecular weights >180 kDa were observed (Fig. 2e, Lane P; Supplementary Data 2). In addition, western blotting detected another band (~80 kDa), possibly representing a Cry78Aa dimer (Fig. 2e, Lane P; Supplementary Data 2). Only Cry78Aa monomeric protein was observed in the supernatant, with bands of ~40 kDa (Fig. 2e, Lane S;

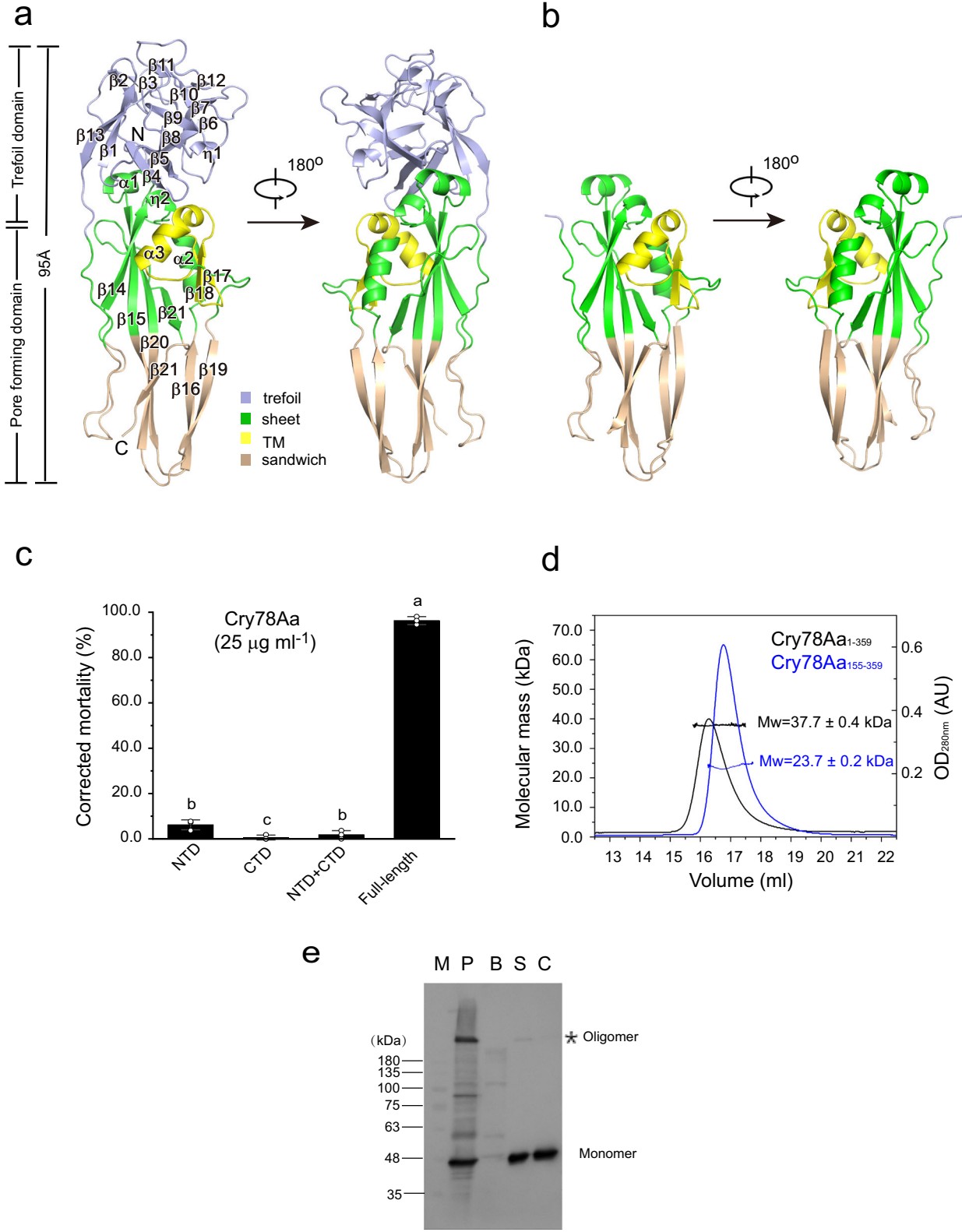

Supplementary Data 2). There have no detectable oligomers in samples of individual BBMV or Cry78Aa treated in same procedure (Fig. 2e, Lane B and Lane C; Supplementary Data 2). These results suggested that the presence of *L. striatellus* BBMVs can trigger the transformation of Cry78Aa from the monomer into oligomers in physiological condition.

**The NTD of Cry78Aa is involved in carbohydrate binding and receptor recognition**. The full-length Cry78Aa has the highest similarity with BinAB and Cry35Ab1 in the overall structure. However, their N-terminal trefoil domains show concrete differences. A structural similarity search in PDB using Cry78Aa residues 1–155 as the probe revealed that the trefoil domain of

**Fig. 2 Structural organization and insecticidal activity of monomeric Cry78Aa. a** Two views of the structure of Cry78Aa$_{13-359}$ (Mol A in Fig. 1a) shown as a ribbon cartoon. Monomeric Cry78Aa$_{13-359}$ has a rod-like shape, ~95 Å in length and 30 Å in diameter, comprising 21 β-sheets and 3 α-helices. All helices and sheets are indicated by numbers on the ribbons (see also Fig. s2-f). Cry78Aa$_{13-359}$ consists of an N-terminal trefoil domain (light blue) and a C-terminal β-pore-forming domain. The pore-forming domain can be further divided into the sheet domain (green), the transmembrane (TM) domain (yellow) and the sandwich domain (brown). **b** Cartoon representation of monomeric Cry78$_{155-359}$ (Mol B) from two perspectives. The color used for each subdomain of Cry78Aa$_{155-359}$ is identical with that in **a**. **c** Insecticidal activity of Cry78Aa against *L. striatellus* nymphs. The individual NTD or CTD of Cry78Aa had no detectable activity against third-instar nymphs at 25 μg ml$^{-1}$. Mixed NTD and CTD did not induce death of the nymphs. There was no significant difference in the death of test nymphs in these treatments compared with the buffer control. Full-length Cry78Aa killed almost all the test nymphs under the same conditions. Bars represent the mean ± SD ($n = 3$ independent experiments). Statistical significance was tested using One-way ANOVA, and is indicated in figures by letters (a, b, c, $P < 0.05$). The difference is not significant when it contains the same letters, and it is significant only when the letters are completely different. **d** Static light-scattering results of the full-length Cry78Aa and the pore-forming domain Cry78Aa$_{155-359}$. The black and blue lines represent the elution profiles of Cry78Aa$_{1-359}$ and Cry78Aa$_{155-359}$ in gel filtration, respectively. The calculated molecular weights (MWs) are labeled accordingly and the error represents the standard deviation of two independent experiments. The X-axis represents the elution volume of the samples, while the Y-axis on the left indicates the scales of the MW, and the Y-axis on the right indicates the UV absorbance of the samples at 280 nm. **e** The BBMVs of *L. striatellus* promoted Cry78Aa oligomer formation. Lane M: Molecular weight marker. Lane P: Pellet obtained after incubation of the Cry78Aa protein with BBMVs. Lane B: BBMVs incubated without Cry78Aa. Lane S: Supernatant obtained after incubation of Cry78Aa with BBMVs diluted 50 times. Lane C: Cry78Aa protein incubated without BBMVs, containing 60 ng of Cry78Aa protein. The immunoblots are representative of three independent experiments, the original images of immunoblots are shown in Supplementary Fig. 8.

Cry78Aa shares the highest identity with that of lectin family proteins, including ricin B, hemoagglutinin (HA33, HA1, HA3), exo-β-1,3-galactanase, and mosquitocidal toxin (MTX) (Supplementary Table 2). As lectins can directly bind different kinds of carbohydrates, these results suggest that the NTD of Cry78Aa could potentially bind carbohydrate or sugars. The proteins in the lectin family are divided into five subfamilies, based on their binding activities against distinct carbohydrates. As no suspected electron density of carbohydrate was observed adjacent to the structure of the trefoil domain of Cry78Aa, the analog of carbohydrate that Cry78Aa binds remains unclear.

To investigate which carbohydrate analog Cry78Aa could bind to, phylogenetic analysis was employed. We chose the top four hits of the Dali search results, the NTD trefoil domains of Cry78Aa, BinA, BinB, and Cry35Ab1 to draw the phylogenetic tree plot. Although the trefoil domains of BinAB and Cry35Ab1 were not among the highest hits in the structure-based search, they are covalently linked to aerolysin-type pore-forming domains and could have more functional relevance to the NTD of Cry78Aa. The phylogenetic tree and sequence alignment results revealed that the trefoil domain of Cry78Aa has the highest genetic similarity with exo-β-1,3-galactanase from *Clostridium thermocellum* (PDB: 3VSF), which contains a carbohydrate-binding module also known as the ricin B lectin domain[17] (Fig. 3; Supplementary Fig. 6a). The ricin B lectin belongs to the S-type subfamily and is reported to specifically bind sugar molecules comprising β-galactoside bonds, including lactose, galactose, and IPTG[18–20] (Supplementary Fig. 6b, c). Thus, we hypothesize that the trefoil domain of Cry78Aa may also bind sugars containing β-galactoside bond as well. Isothermal titration calorimetry (ITC) experiments confirmed that Cry78Aa indeed bound to galactose, GalNAc and lactose, and the fitted binding dissociation constants ($K_d$) were approximately 15.5, 9.8, and 10.2 mM, respectively (Supplementary Fig. 7a–c). Meanwhile, the other carbohydrates, such as glucose, GlcNAc, mannose and arabinose, had no detectable interaction with Cry78Aa (Supplementary Fig. 7d–g).

The N-terminal trefoil domain of Cry78Aa can be divided into two layers from a side view: the cap layer and the barrel layer. It can also be divided into three blades from the top or bottom view (Fig. 4a). A structural alignment of Cry78Aa-NTD and sugar-binding ricin B indicated that the trefoil domain of Cry78Aa adopts an almost identical conformation with galactose binding ricin B (PDB: 3VT1)[17], with an RMSD (over 165 Cα atoms of the two structures) of 0.41 Å (Fig. 4b). In addition, the main chain adjacent to the galactose binding position superposed better than

the other regions. As observed in the structure of galactose-bound ricin B, four residues, D416, Y431, N438, and Q439, could contribute most to galactose recognition and binding. This was consistent with a previously reported NQxF motif in lectins that are involved in sugar binding. The four residues are conserved in the trefoil domain of Cry78Aa, corresponding to D86, Y100, N107, and Q108, and their side chains completely superimposed with that of ricin B (Fig. 4b). These results suggest that the blade of the trefoil domain of Cry78Aa is capable of binding or accommodating one galactose molecule. Notably, 3 blades of the trefoil domain of Cry78Aa share normal similarity in side chain orientation and amino sequence, and the putative sugar-binding residues are identical (Fig. 4c, d). These results imply that the NTD of Cry78Aa contains multiple candidate sugar binding sites, which may function to bind glycolipids or glycoproteins on the surface of the cell membrane. In addition to galactose, the blades on the trefoil domain of Cry78Aa may also bind other sugars. Structural alignment of the NTD of Cry78Aa and IPTG-bound (PDB: 3VT2) or multigalactose-bound ricin B (PDB: 3VSZ)[17] revealed that the conserved residues of Cry78Aa also share an identical orientation with those of ricin B (Supplementary Fig. 6b, c).

To investigate the role of the putative sugar binding site on Cry78Aa, we mutated the key residues that potentially recognize the sugar and subsequently tested the insecticidal activity of the mutant proteins. As expected, the mutants containing a single residue mutation on blade 2 (including D86A, Y100A, N107A, or Q108A) showed significantly decreased insecticidal activity against the third-instar nymphs of *L. striatellus* compared to the wild Cry78Aa. Specifically, under the 25 μg ml$^{-1}$ protein treatment, the mortality rate of the nymphs fed mutant proteins was lower than that of the nymphs fed wild-type proteins (Fig. 4e; Supplementary Data 1). Among these Cry78Aa mutants, N107A showed the most significant impact on its insecticidal activity and the effects of mutants bearing Y100A, D86A, and Q108A were gradually attenuated. In addition, the individual mutations of D39A, Y54A, N61A, or Q62A on blade 1, and D128A, Y143A, N150A, or Q151A on blade 2 of the trefoil domain can reduce the toxicity of Cry78Aa (Fig. 4e; Supplementary Data 1). Although the impacts of mutants on blade 1 and blade 3 differed slightly compared to the mutations on blade 2, an overall reduction in insecticidal activity was consistently observed. Some mutations on the trefoil domain considered irrelevant to sugar binding, such as D47A and N115A, are included as negative controls. As expected, these mutants showed no obvious influence on the insecticidal activity of Cry78Aa (Fig. 4e; Supplementary Data 1). During the

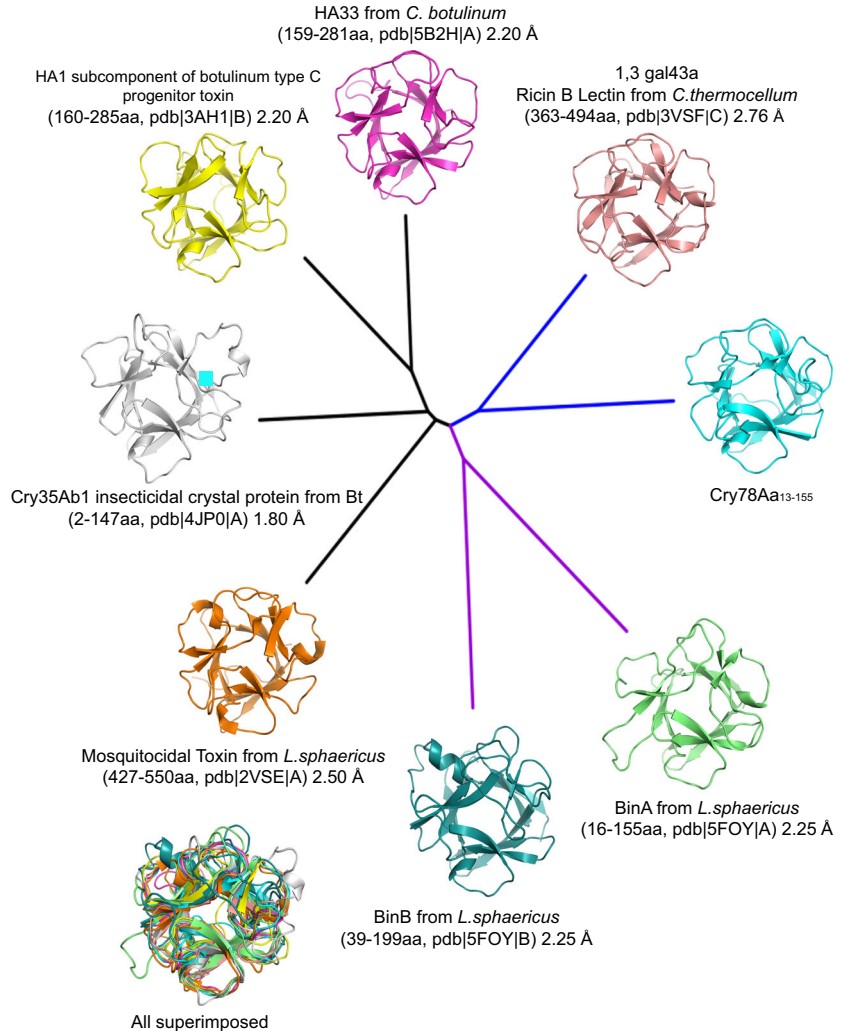

**Fig. 3 Structural relationships of the NTD of Cry78Aa with other trefoil domains illustrated by a phylogenetic tree plot.** Four of the structures used for comparison were identified from a structural similarity search through the Protein Data Bank (PDB) conducted by the Dali server (using Cry78Aa residues 13–155 as the probe, see also Supplementary Table 2). The top four hits included HA33 from *C. botulinum* (PDB: 5B2H), the HA1 subcomponent of botulinum type C toxin (PDB: 3AH1), ricin B lectin from *C. thermocellum* (PDB: 3VSF) and mosquitocidal toxin from *L. aphaericus* (PDB: 2VSE). The remaining structures chosen for comparison, including Cry35Ab1 from *B. thuringiensis* (PDB: 4JP0), BinA, and BinB (PDB: 5FOY), were selected based on their overall structural similarity with Cry78Aa. That is, they all consisted of a trefoil domain covalently linked to a β-pore-forming domain. The phylogenetic tree was generated by using MEGA-X (10.0.2) and the input sequences of the relevant structures were downloaded from the PDB. The selected trefoil domains, including Cry78Aa, can be divided into 3 clades, which are indicated by black, blue, and purple lines. The NTD of Cry78Aa has the closest homology with ricin B lectin and was assigned to the same clade (both marked in blue). Superposition of all the trefoil domains exhibited an overall similar conformation.

expression and purification procedures, all the mutated Cry78Aa proteins exhibited good yield and behavior, implying that they were well folded. These results confirmed that the carbohydrate-binding site of Cry78Aa is essential for its toxicity to rice planthoppers. To further elucidate the carbohydrate analog on the cell surface that potentially affect the activity of Cry78Aa, we performed a sugar protection assay to assess the effects of Cry78Aa against *L. striatellus*. Feeding third-instar nymphs a high concentration of galactose protected the nymphs from death induced by wild Cry78Aa. Approximately 65% of the tested nymphs died in 7 days when fed $15 \mu g \, ml^{-1}$ wild Cry78Aa in forage alone (Fig. 4f; Supplementary Data 1). In contrast, the addition of an extra $30 \, mg \, ml^{-1}$ (mass ratio of 2000:1) galactose in the same treatment rescued the nymphs, where nearly 30% of the total nymphs tested died during the assessment (Fig. 4f; Supplementary Data 1). These results indicated that the addition of external galactose significantly affected the activity of Cry78Aa

and that galactose conjugated to receptors located at the cell membrane is probably the target of Cry78Aa in vivo. We also tested the effects of some other kinds of sugars including arabinose, lactose, GalNAc, glucose, GlcNAc, and mannose which are universally found in insect cell membranes. However, only lactose and GalNAc exerted a protective effect on nymph comparable to that of galactose. The effects of other sugars on the activity of Cry78Aa were not as significant as those of galactose, as more nymphs survived under the same conditions (Fig. 4f; Supplementary Data 1). These findings were consistent with previous ITC binding results and confirmed the significance of galactose derivatives in recognizing the trefoil domain of Cry78Aa in vivo.

**The CTD of Cry78Aa resembles that of aerolysin family toxin.** The CTD of Cry78Aa is regarded as the pore-forming domain. To identify more functionally related proteins, we ran a Dali search in

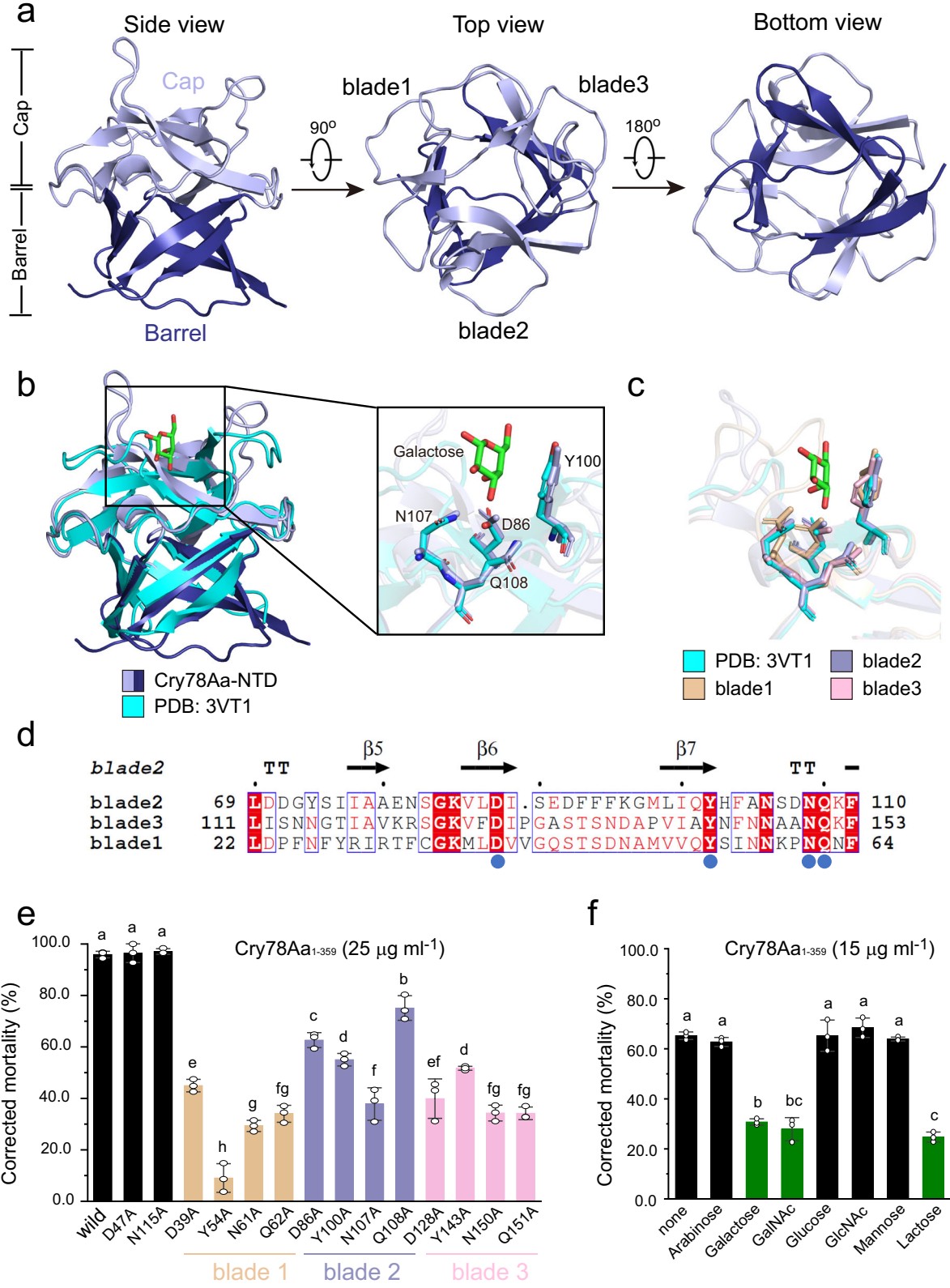

the PDB database using the structure of Cry78Aa$_{155-359}$ as the search model. The top three hits showing structural similarity to Cry78Aa$_{155-359}$ were BinA, Cry35Ab1, and BinB (Supplementary Table 3). This was similar to the Dali search results using Cry78Aa$_{13-359}$ as coordinates. Other less similar yet comparable structures comprised several members of the well-known aerolysin family toxin, including lysenin[21], hemolytic lectin[22], parasporin[14],

epsilon toxin[23,24], monalysin[25,26], and proaerolysin[27] (Supplementary Table 3). Therefore, the pore-forming domain of Cry78Aa may belong to the aerolysin family.

A sequence alignment of Cry78Aa$_{155-359}$ and the aerolysin-type toxins revealed that it shares low identity with other aerolysin members. However, Cry78Aa$_{155-359}$ has several characteristic features of aerolysin-type pore-forming toxins. Cry78Aa$_{155-359}$

**Fig. 4 The NTD trefoil domain of Cry78Aa may bind galactose. a** Views of the NTD of Cry78Aa from different perspectives. The side view (left diagram) of the NTD shows that it consists of two layers, the cap layer (in light blue) and the barrel layer (in blue), which have nearly identical lengths. The top view (middle diagram) is shown from the cap side. It can be seen clearly that the trefoil domain comprise three blades, which are indicated beside. The bottom view (right diagram) was observed from the barrel side. **b** Structural alignment of the trefoil domain of Cry78Aa and the galactose-bound ricin B lectin domain (PDB: 3VT1). The color representation of Cry78Aa was the same as in **a**. The ricin B protein moiety was shown in cyan, while the carbon and oxygen atoms of the galactose molecule are shown in green and red, respectively. The smaller frame in the left diagram marks the galactose binding pocket of ricin B, while the larger frame in the right diagram represents the close-up view. The conserved resides of Cry78Aa (on blade 2) and ricin B that may interact with galactose are marked and shown as sticks (color representations: carbon, light blue for Cry78Aa and cyan for ricin B; nitrogen, blue; oxygen, red). **c** Structural alignment of three blades of the trefoil domain and ricin B. **d** Sequence alignment of three blades of the Cry78Aa trefoil domain. Secondary structural units of blade 2 are shown on the top. The numbers flanking the sequence specify the start and end of each blade. Totally conserved residues in the three blades are marked by a red background. Blue solid circles below represent the residues of Cry78Aa that were predicted to be involved in galactose binding. **e** Effects of the mutants on the insecticidal activity of Cry78Aa proteins against *L. striatellus* nymphs. Experiments were performed as described in the methods. Each column represents the average value of two independent trials, and error bars represent the standard deviation. The results of mutants on each blade have the same color representation as that in **c** for clarity. Two irrelevant mutants of Cry78Aa, D47A, and N115A (see also Supplementary Fig. 5c), were tested as negative controls. **f** Effects of sugars on the insecticidal activity of Cry78Aa against *L. striatellus* nymphs. Assays were performed as described in the methods, and the results are presented as in **e**. The results of the assay after treatment with galactose, GalNAc, and lactose are shown in grass green. Bars represent the mean ± SD ($n = 3$ independent experiments) in **e**, **f**. Statistical significance was tested using One-way ANOVA, and is indicated in figures by letters (a, b, c, ..., $P < 0.05$). The difference is not significant when it contains the same letters, and it is significant only when the letters are completely different.

have conserved building blocks possessed by the aerolysin-type pore-forming domain. The conserved secondary structural units comprise at least five long twisted β-sheets that swap with one another and a putative membrane-spanning region (MSR) that resides between β-sheets 2 and 3 (Fig. 5). The putative MSR from Cry78Aa, BinAB, and Cry35Ab1 consists of an α-helix and a pair of short β-sheets, whereas the promonalysin MSR contains two α-helixes, and the MSR in the other members of the aerolysin family toxin mentioned above comprise mainly β-sheets. Consequently, Cry78Aa may use its pore-forming domain to form a giant complex to drill holes on the cell membrane, similar to aerolysin family toxins.

The putative MSR of Cry78Aa consisted of alternating hydrophobic and hydrophilic amino acid residues (Fig. 6a, b). This feature attributes the MSR of Cry78Aa an amphiphilic property which is commonly shared among aerolysin family toxins. There also existed serine/threonine patches in the CTD of Cry78Aa (Fig. 6c), similar to aerolysin-type pore-forming toxins. Sequence analysis showed that 20.5% of Cry78Aa$_{155-359}$ consisted of either serine (7.8%) or threonine (12.7%). These serine/threonine residues were mainly distributed inside or in vicinity of the MSR (Supplementary Fig. 3f), constructed a hydrogen bond network by interacting with each other or with other nearby residues (Fig. 6d). Although explicit function of the serine/threonine patches remains to be investigated, we proposed they could act like a lock to stabilize the amphiphilic MSR under untriggered conditions and help MSR transformed and extended to the membrane in appropriate environment.

**The mechanism by which Cry78Aa uses to kill rice planthoppers**. There are several essential steps for β-pore-forming toxins to exert their activity. One of these steps prior to the membrane hole-drilling process is the oligomerization of the toxin, which could be triggered by alkaline surroundings, limited proteolysis, and receptor binding on the cell surface. Due to the lack of the Cry78Aa-receptor complex structure and its corresponding oligomer during the pore state, we can only propose a schematic model in which Cry78Aa recognizes its receptor and changes from a monomeric inactive state to the pore state (Fig. 7). As we proposed that the trefoil domain of Cry78Aa can bind galactose, GalNAc or lactose, it is possible that Cry78Aa can recognize the carbohydrate conjugately linked to lipids or proteins on the cell surface by its NTD. With the help of receptors, the pore-forming domains of Cry78Aa assemble from the lateral side to form a

giant oligomer. The physiological conditions in the midgut of nymphs, such as pH or the presence of special kinds of proteases, may also facilitate the process. The specificity (regarding the type of nymph Cry78Aa kills) and efficiency of Cry78Aa are mainly attributed to the existence or distribution of galactose-linked receptors on the cell membrane of the midgut.

Another crucial step for the functioning of Cry78Aa is the molecular rearrangement of each protomer inside the oligomer. Considering that the β-pore-forming domain of Cry78Aa shares conserved structural features with other aerolysin family toxins, this process can be deduced from the case of recently reported aerolysin (Fig. 7). Aerolysin undergoes at least four stages, including prepore, post-prepore, quasipore, and mature membrane-inserted pore states during the drilling process. It employs a swirling mechanism to promote the transition from the prepore to pore conformation[21,28–30]. In this step, the relative position of the NTD and CTD of aerolysin changes dramatically, and its pre-stem loop is released from beside the consensus β-sheets. This leads to the formation of a membrane-spanning the β-barrel structure, which comprises seven pairs of β-sheets. Upon receptor binding or triggering by other undetermined effects, Cry78Aa protomers in the oligomer could also experience a conformational change similar to that of aerolysin. The putative MSR of Cry78Aa, which consists of an α-helix and two short β-sheets, transforms into a pair of long antiparallel β-sheets, and subsequently forms a β-barrel. Thus, the oligomer of Cry78Aa transforms from an inactive prepore state to an active pore state. This represents the proposed process in which the oligomer of Cry78Aa transforms from the inactive prepore state to the active pore state.

The observation in this study, including the trefoil domain of Cry78Aa bind sugar such as galactose, lactose, or GalNAc, and the existence of a direct interaction between the NTD and CTD of Cry78Aa sheds light on the mechanism by which Cry78Aa kills rice planthoppers. However, more investigation is needed to fully reveal the integral progress.

## Discussion
Although Cry and Vip family genes have been universally used in genetic engineering to control injurious insects in green agricultural production, the gene source reservoir for potential application is rather poor at present, especially that with independent intellectual property rights in China. In addition, rice planthoppers use piercing and sucking mouthparts to feed and transmit poison, which induces dramatic disease and decreases

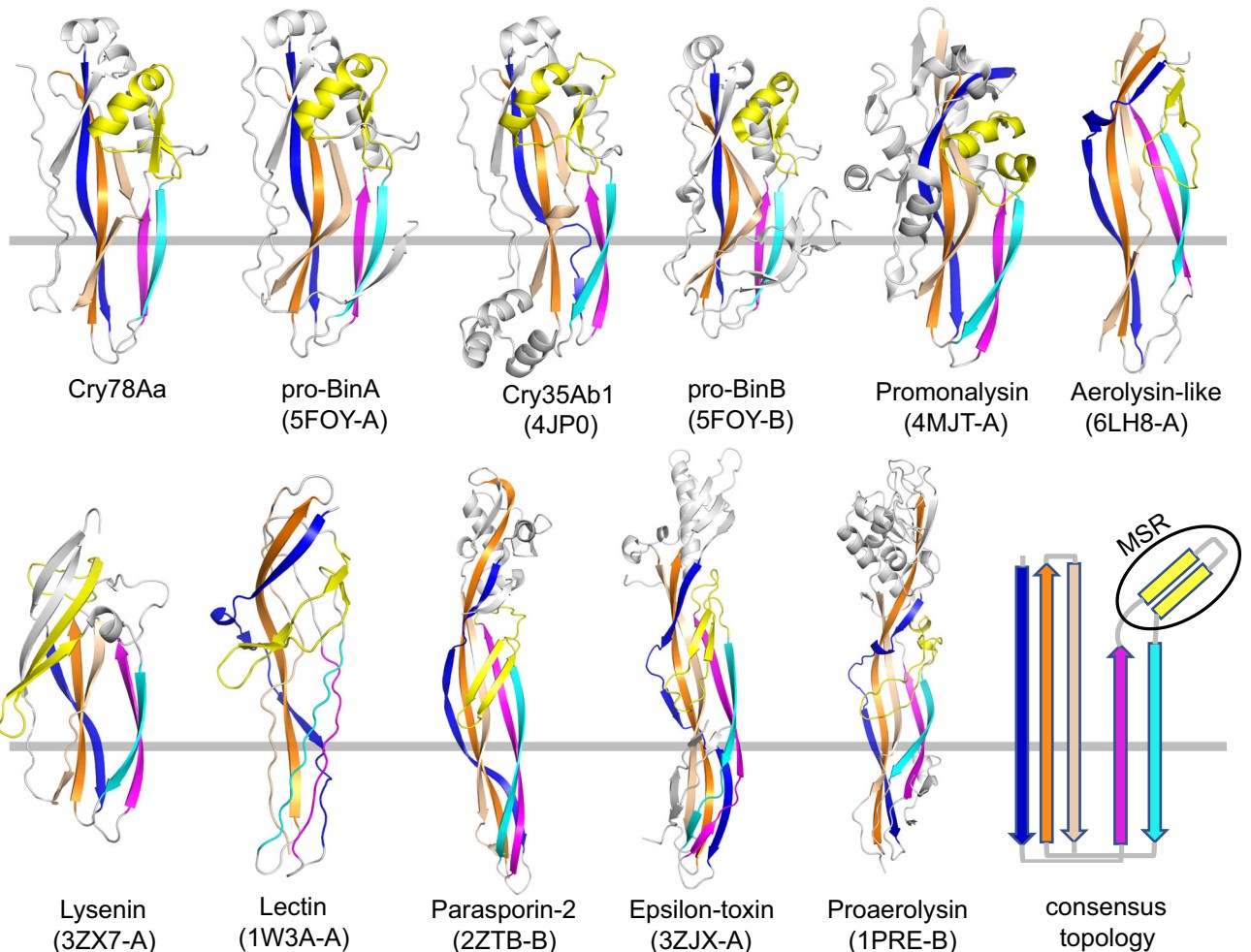

**Fig. 5 The CTD pore-forming domain of Cry78Aa resembles those of aerolysin family toxins.** Topology of the aerolysin family of β-pore-forming toxins selected from Dali search results using Cry78Aa$_{155-359}$ as a model (see Supplementary Table 3). These proteins share a core topology composed of five antiparallel β-sheets and a putative membrane-spanning region (MSR, in yellow). PDB codes are included in parentheses. Any accessory domains outside the pore-forming module (PFM) of the toxins were excluded for clarity. The PFM was divided into two subdomains: a β-sheet subdomain at one side (above the horizontal gray line) and a β-sandwich subdomain at the opposite side (below the horizontal gray line). The secondary structure, including length, twist, number of β-sheets, and putative membrane-spanning region (in yellow), varies widely among the selected toxins. However, in all cases the putative MSR region is located between the second and third sheets, suggesting that these toxins might share a common mechanism of pore formation.

the production of crops. In the long run, transgenic Bt rice is the most effective control method. Cry78Aa could be a useful candidate gene in genetic engineering to battle rice planthoppers since it has a remarkable effect on the survival of nymphs in vitro. The advantages of Cry78Aa in application includes at least two aspects. First, the structure of monomeric Cry78Aa is simple and its molecular weight is not high. Second, Cry78Aa itself which does not require assistance from any other partners, is sufficient to exert the entire activity. Many binary toxins, such as BinA/BinB, Cry64Ba/Cry64Ca, and Cry34Ab1/Cry35Ab1, work efficiently only when both components are present[4,9,10]. These features make Cry78Aa convenient for directed evolution to achieve engineered toxins with higher activity in vitro or for further application for genetic improvement in vivo. The structure we resolved in this study lays a foundation for structure-based toxin design. The bioassay performed above can also provide some clues for how to obtain stronger engineered Cry78Aa. Moreover, the concrete molecular mechanism of each step in progress of Cry78Aa intrude into cell, which needs to be further clarified, will provide more solid instruction.

There is a concern about whether binding of the trefoil domain of Cry78Aa to carbohydrates on the surface of the cell membrane is enough to trigger oligomerization of Cry78Aa and the subsequent drastic conformational change of each protomer, which drives the drilling process. We observed that there are three putative carbohydrate-binding sites, which share identical amino acid categories involved in recognizing sugars, on the blades of the trefoil domain of Cry78Aa, implying that one Cry78Aa molecule may bind with several sugar molecules. Although the trefoil domain of Cry78Aa has strong potential to bind with galactose, it does not exclude the possibility that other kinds of sugars can also be recognized by Cry78Aa. Structural alignment of the trefoil domain of Cry78Aa and ricin B lectin suggested that the former used its second blade to interact with galactose, IPTG, or other kinds of sugar containing galactoside. In addition, Cry78Aa may recognize sugars by using components other than the trefoil domain. It is been reported that some kinds of lectins use the bottom face of the trefoil domain to bind with carbohydrates. Given the structural similarity between trefoil domain of Cry78Aa with lectins, it is possible that Cry78Aa associates with sugars by an analogous strategy. These effects could collectively and partially contribute to the formation and transition of the pore state.

The binding of Cry78Aa with glycolipids or glycoproteins on the cell surface may represent only the preliminary step in the

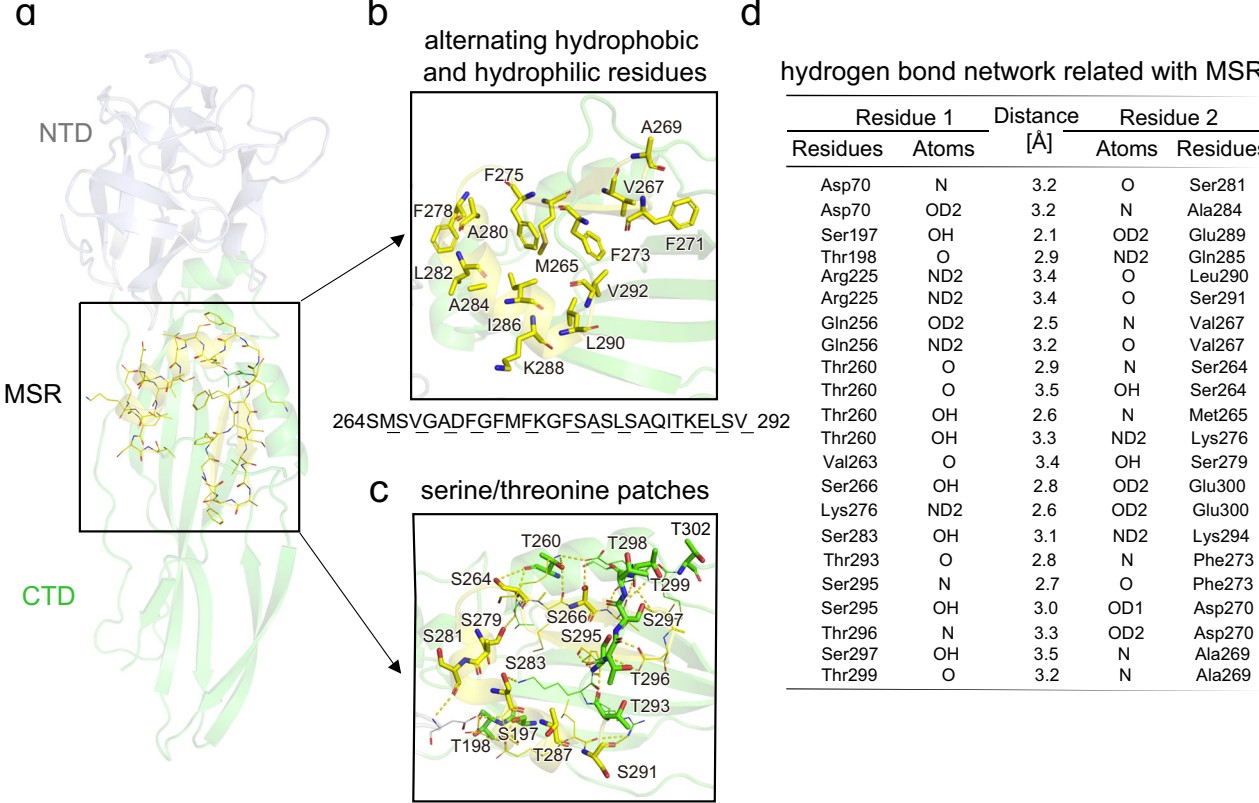

**Fig. 6 Cry78Aa share common characteristics of aerolysin-type pore-forming toxins. a** The putative membrane-spanning region (MSR). The NTD and CTD are shown in color of light blue and green, respectively. The MSR located at the CTD and is enclosed in a black frame, colored in yellow, and the residues shown as lines (color representations: carbon, yellow; nitrogen, blue; oxygen, red; sulfur, orange). The ribbon representation is shown in partial transparency for clarity. **b** Close-up view of the alternating hydrophobic and hydrophilic residues in the MSR. The hydrophobic residues are shown as sticks and labeled. The hydrophilic residues are omitted for clarity. Amino acids sequence of the MSR is listed below and the hydrophobic residues are underlined. Color representation is the same as diagram (**a**). **c** Close-up view of the serine/threonine patches intra or in vicinity of the MSR. All the hydrogen bond interactions intra MSR or between MSR and other residues are presented. The yellow dashed lines represent hydrogen bonds. Serine and threonine participate in the interactions are shown as sticks, with category and number labeled beside. Other residues participate in the interactions are shown as lines (color representations: carbon, yellow for MSR and green/light blue for CTD/NTD; nitrogen, blue; oxygen, red; sulfur, orange). **d** Detailed depiction of the hydrogen bonds presented in diagram (**b**). Most hydrogen bonds consisted of residues that occurred in serine/threonine patches.

process of function. Generally, the binding energy of sugars with their partner is not high enough to drive the subsequent conformational change of the target. In the case of Cry78Aa, lipids or proteins conjugated with galactose or other kinds of sugar are more likely to provide sites for the trefoil domain of Cry78Aa to dock. The interaction could be rather rigid, and it is difficult to trigger the drastic rearrangement of Cry78Aa molecules in the oligomeric state. Thus, receptors other than glycoproteins or glycolipids on the membrane could also be essential for the effects of Cry78Aa on rice planthoppers. Considering that Cry78Aa selectively kills planthopper nymphs, the membrane component of the midgut cells should play a decisive role.

The pore-forming domain of Cry78Aa executes the drilling action. In contrast to cell membrane adsorption and docking by the trefoil domain, oligomerization and pore state transition directly involve the pore-forming domain of Cry78Aa. Much of the aerolysin protein family has been investigated in both monomeric and pore states, shedding light on the mechanism by which Cry78Aa functions. Drastic conformational changes in the pore-forming domain, especially the putative membrane-spanning region, may be driven by intermolecular interactions between protomers in the oligomer or other surrounding factors, such as pH and limited proteolysis[21,28,30,31]. The oligomerization and pore state transition of the pore-forming domain of Cry78Aa could resemble those of other aerolysin family members.

Nevertheless, the process by which Cry78Aa invades the cell membrane could differ from those of other pore-forming toxins, given that the pore-forming domain of Cry78Aa is covalently linked to a trefoil domain that is classified as an S-type lectin. The well-folded pore-forming domain of Cry78Aa alone is not enough to trigger the death of planthoppers suggesting the vital role of its trefoil domain in docking on the cell surface or supplying energy for oligomerization. Although the lack of the Cry78Aa pore state prevents us from deducing its exact process, the monomeric structure we resolved here lays a foundation to fully understand its mechanism and paves the way to applying the toxin to combat planthoppers in agricultural production.

We can design mutations in Cry78Aa to engineer proteins with higher pesticidal activity against rice planthoppers based on the resolved structure here, even though the mechanism remains to be fully revealed. In the process of verifying the functions of the key residues on the trefoil domain of Cry78Aa, we obtained several mutants with higher activity than the wild type. This result suggests that the activity of Cry78Aa has the potential to be further improved. Cry78Aa consists of a trefoil domain and a pore-forming domain, both of which are essential for its activity, suggesting that the two components can be designed to increase the efficiency of Cry78Aa. For example, it is possible to design mutations in the pore-forming domain of Cry78Aa, based on sequence and structure comparisons between Cry78Aa and other

### hydrogen bond network related with MSR

| Residue 1 | | Distance [Å] | Residue 2 | |
|---|---|---|---|---|
| Residues | Atoms | | Atoms | Residues |
| Asp70 | N | 3.2 | O | Ser281 |
| Asp70 | OD2 | 3.2 | N | Ala284 |
| Ser197 | OH | 2.1 | OD2 | Glu289 |
| Thr198 | O | 2.9 | ND2 | Gln285 |
| Arg225 | ND2 | 3.4 | O | Leu290 |
| Arg225 | ND2 | 3.4 | O | Ser291 |
| Gln256 | OD2 | 2.5 | N | Val267 |
| Gln256 | ND2 | 3.2 | O | Val267 |
| Thr260 | O | 2.9 | N | Ser264 |
| Thr260 | O | 3.5 | OH | Ser264 |
| Thr260 | OH | 2.6 | N | Met265 |
| Thr260 | OH | 3.3 | ND2 | Lys276 |
| Val263 | O | 3.4 | OH | Ser279 |
| Ser266 | OH | 2.8 | OD2 | Glu300 |
| Lys276 | ND2 | 2.6 | OD2 | Glu300 |
| Ser283 | OH | 3.1 | ND2 | Lys294 |
| Thr293 | O | 2.8 | N | Phe273 |
| Ser295 | N | 2.7 | O | Phe273 |
| Ser295 | OH | 3.0 | OD1 | Asp270 |
| Thr296 | N | 3.3 | OD2 | Asp270 |
| Ser297 | OH | 3.5 | N | Ala269 |
| Thr299 | O | 3.2 | N | Ala269 |

264 SMSVGADFGFMFKGFSASLSAQITKELSV 292

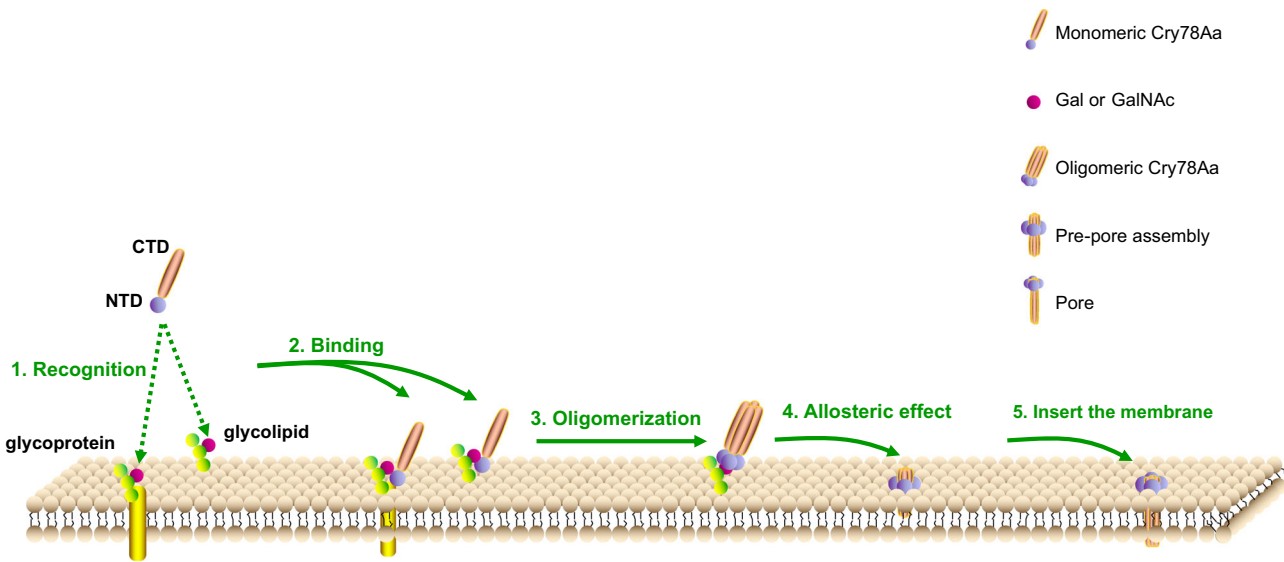

**Fig. 7 The proposed model Cry78Aa uses to kill rice planthoppers.** According to the structural and biochemical results in this study, we proposed that the Cry78Aa may kill *L. striatellus* nymphs by the following steps: I. The NTD of Cry78Aa recognizes carbohydrates such as galactose, lactose or GalNAc on the cell membrane surface receptor. II. The NTD of Cry78Aa binds to receptors (such as Gal-linked glycolipid or glycoprotein) on the membrane under the guidance of carbohydrates. III. With the help of receptors, the local concentration increase of monomers will promote oligomerization, and the pore-forming domains (CTDs) of Cry78Aa assemble to form a giant oligomer. IV. Molecular rearrangement of each protomer inside the Cry78A oligomer, leading to a prepore assembly. V. β-Barrel formation will create a pore in the bilayer, leading to the disruption of membrane homeostasis and ultimately to cell lysis.

effective β-pore-forming toxins to improve its pore-forming ability. More exploration related to mechanism elucidation and protein engineering is needed to maximize the usage of this promising toxin in agriculture.

## Methods

**Molecular cloning, protein expression, and purification**. Full-length *cry78aa* gene was amplified from the Bt strain C9F1 and subcloned into pET15D (Novagen) that fused with a 6 × His tag at the N-terminus of expressed fusion protein. The plasmid was transformed into BL21 (DE3) and 1 liter of lysogeny broth medium supplemented with 100 μg ml$^{-1}$ ampicillin was inoculated with a transformed bacterial pre-culture and shaken at 37 °C until the optical density at 600 nm reached 1.0. After being induced with 0.2 mM isopropyl-β-D-thiogalactoside and growing at 16 °C for 16 h, the bacterial pellet was collected and homogenized (JNBIO, China) in a buffer containing 25 mM Tris-HCl, pH 8.0, and 150 mM NaCl. After centrifugation at 23,000 × *g* at 4 °C, the supernatant was loaded onto a column equipped with Ni$^{2+}$ affinity resin (Ni-NTA, Qiagen); washed with a buffer containing 25 mM Tris-HCl, pH 8.0, 150 mM NaCl, and 15 mM imidazole; and eluted with a buffer containing 25 mM Tris-HCl, pH 8.0, 150 mM NaCl, and 250 mM imidazole. The eluted protein was applied to a 7 ml Source-Q10/100 column (GE Healthcare), followed by a gradient NaCl elution (up to 1 M) in 25 mM Tris-HCl, pH 8.0. The elution peak was concentrated to 1 ml (~20 mg ml$^{-1}$) and digested by drICE (0.1 mg ml$^{-1}$) at 4 °C for 1 h without stopping digestion before being subjected to gel filtration chromatography (Superdex-200 Increase 10/300, GE Healthcare) which was equilibrated with a buffer containing 25 mM Tris-HCl, pH 8.0, 150 mM NaCl and 5 mM 1,4-dithiothreitol (DTT). The peak fractions were collected for crystallization trials. Mutants of *cry78aa* were constructed by overlap PCR and subcloned into pET15D as well. Purifications of the mutants or boundaries of Cry78Aa were same as that of the full-length protein.

**Crystallization**. Cry78Aa$_{13-359}$ was crystallized by using the hanging-drop vapor-diffusion method at 18 °C, and 1 μl of sample was mixed with an equal volume of reservoir solution. Diamond-shaped crystals appeared overnight from a reservoir solution containing 9.5% PEG3350, 150 mM NH$_4$Cl, 0.1 M Tris-HCl, pH 7.6. After 48 h of growth, the crystal ceased growing and was flash frozen in liquid nitrogen. The cryoprotection buffer included 20% ethylene glycol added to reservoir solution. The crystal of Cry78Aa$_{13-359}$ diffracted beyond 2.9 Å at the Shanghai Synchrotron Radiation Facility (SSRF) beamline BL17U1[32]. Cry78Aa$_{155-359}$ was crystalized in a solution containing 21.4% PEG3350, 0.2 M Mg (NO$_3$)$_2$, 0.1 M Bis Tris-HCl, pH 6.0 and diffracted 2.4 Å at the SSRF beamline BL17U1.

**Data collection and structure determination**. The dataset was collected at the SSRF beamline BL17U1 and processed with the HKL3000 or HKL2000 package[33].

Further processing was performed with the CCP4 suite[34]. Data collection and structural refinement statistics are summarized in Table 1. The structure of Cry78Aa$_{13-359}$ was solved by molecular replacement (MR) using the coordinate of Cry78Aa$_{155-359}$, whose structure was solved by single-wavelength anomalous diffraction (SAD) method by grow derivative crystal with selenomethionine. The structure was manually and iteratively refined with PHENIX[35] and COOT[36]. All figures representing structures were prepared with PyMOL (http://www.pymol.org).

**Static light scattering experiment**. Approximately 1 mg ml$^{-1}$ Cry78Aa$_{13-359}$ and Cry78Aa$_{155-359}$ was independently loaded onto a Superdex 200 10/300 column connected to a HELEOS multi-angle light scattering instrument (WYATT Technology). The proteins were eluted with 25 mM Tris buffer, pH 8.0, and 150 mM NaCl at a flow rate of 0.5 ml min$^{-1}$. Each fraction was automatically analyzed by using multi-angle light scattering. The figure was drawn using Origin 8.0.

**Bioassay**. The artificial liquid feed for *Laodelphax striatellus* was prepared according to the method of Fu et al.[37] and frozen at −20 °C. The insecticidal activity of each treatment against *L. striatellus* was determined using a double parafilm membrane feeding system. The specific method refers to the method improved by Liu et al.[6]. Each bottle contains 21 third-instar nymphs of *L. striatellus*, and each treatment was repeated three times. Liquid food is changed every 2 days. After feeding continuously for 6 days under the condition of a light:dark photoperiod of 16:8 h and 70% relative humidity at 28 °C, dead insects were counted. The Cry78Aa mutant protein was added to a liquid artificial diet at a final concentration of 25 μg ml$^{-1}$. Protein buffer (20 mM Tris-HCl, pH 8.0) was used as a negative control.

For testing the effect of carbohydrates on the insecticidal toxicity of Cry78Aa (15 μg ml$^{-1}$), protein and carbohydrates were mixed and added to the feed at a mass ratio of 1:2000. The protein buffer was added with carbohydrates of the same quality as the treatment group as a negative control. Carbohydrates are prepared with protein buffer.

**Oligomerization assays with BBMVs**. The midgut of 3rd instar *L. striatellus* nymphs was used for BBMV isolation by the differential precipitation method using MgCl$_2$ as reported[38], and total BBMVs protein was quantified using the Bradford method. The oligomerization assays with BBMVs were performed according to Ocelotl et al.[39] with minor modification. 400 nM Cry78Aa was added to a mixture of 10 μg BBMVs in binding buffer (0.1% BSA, 0.1% Tween 20 in phosphate-buffered saline [PBS]) in a final volume of 100 μL and incubated at 37 °C for 1 h. The sample containing only BBMV and Cry78Aa incubated in the absence of BBMV were used as controls. Pellets containing the BBMV were recovered by centrifugation at 14,000 × *g* for 10 min at 4 °C, while the supernatant containing unbound protein was recovered and stored. Pellets containing the BBMV were washed twice with 500 μL ice-cold binding buffer. The final BBMV

pellets were resuspended in Laemmli sample buffer. The proteins were separated by SDS-PAGE and electrotransferred onto polyvinylidene difluoride (PVDF) western blotting membrane. After western blot, Cry78Aa was detected with monoclonal mouse anti-Bt Cry78Aa (1:8000, 1 h) from Genecreate followed by secondary antibody (1:10,000, 1 h) coupled with horseradish peroxidase (HRP). Then, Cry78Aa was visualized by chemiluminescence using SuperSignal$^{TM}$ West Pico PLUS kit using an ImageQuant LAS400 image analyzer. The molecular weight marker used was ColorMixed Protein Marker PR1910. The experiment was repeated at least three times.

**Isothermal titration calorimetry.** Cry78Aa protein was analyzed on MicroCal iTC200 instrument (GE Healthcare) in 20 mM Tris-HCl (pH 8.0). Monosaccharide and its derivatives (same buffer as protein) at a concentration of 50 mM were injected into the cell loaded with protein (150 μM, ~200 μL). Titration experiments consisted of 20 injections at 25 °C, and samples were stirred at a speed of 500 rpm. The first injection (0.2 μL) had a duration time of 0.4 s and all subsequent injections (2 μL) were of a duration of 4 s with 150 s between each injection. Origin 7.0 (OriginLab Corp., Northampton, MA, USA) was operated to carry out the raw data processing and analysis, according to the one-site binding model. The heats that each sugar titrated into the buffer under same conditions was subtracted as control.

**Statistics and reproducibility.** Origin version 2021 was used to plot and represent data. All experiments were repeated at least three times. All experimental values were expressed as the mean ± SD, and individual data points are plotted as symbols above the bars of the histograms. Statistical significance was tested using One-way ANOVA, and is indicated in figures by letters (a, b, c, …, $P < 0.05$). The difference is not significant when it contains the same letters, and it is significant only when the letters are completely different.

**Reporting summary.** Further information on research design is available in the Nature Research Reporting Summary linked to this article.

# Data availability

The coordinates and the structure factors have been deposited to the Protein Data Bank with the accession code: 7Y78 and 7Y79. The source data behind the graphs presented in the main figures of the paper can be found in Supplementary Data 1. Uncropped western blots for Fig. 2e can be found in Supplementary Data 2. All other data are available from the corresponding author on reasonable request.

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

# Acknowledgements

We thank the staffs of the BL17U1/BL19U1 beamline of the National Center for Protein Sciences Shanghai (NCPSS) at the Shanghai Synchrotron Radiation Facility for assistance during data collection, and research associates at the Center for Protein Research, Huazhong Agricultural University, for technical support. We thank Dr. Chen Wang for critical reading of the manuscript. This work was supported by the National Key R&D

Program of China (2017YFD0200400), the National Natural Science Foundation of China [31770878 to D.Z.], the open funds of the State Key Laboratory for Biology of Plant Diseases and Insect Pests [SKLOF201904], the open funds of the National Key Laboratory of Crop Genetic Improvement [ZK201907], the open funds of the State Key Laboratory of Hybrid Rice [KF201703], and the Huazhong Agricultural University Scientific & Technological Self-innovation Foundation [2662017QD034, 2662018JC023].

## Author contributions

J.Z. and D.Z. designed and supervised this study. B.C. and Z.W. performed bioassay analysis. Y.N., C.C., and N.W. performed the protein purification, crystallization, and biochemical experiments. Z.G. and B.C. solved the structure. B.C., C.S., J.Z., and D.Z. prepared the manuscript with inputs from other authors. All authors read and commented on the manuscript.

## Competing interests

The authors declare no competing interests.
