## [Peer Review File · Communications Biology]

Reviewers' comments:

Reviewer #1 (Remarks to the Author):

COMMSBIO-21-1677-T

This manuscript reports the first crystal structure of Cry78Aa consisting of two domains with a similar structural folding to that of aerolysin-like family. The structural data also suggest a mechanism underlying the function of Cry78Aa against rice planthopper and provide useful guidelines for further engineering of protein for maximizing the toxicity.

Below are some comments for this article:

1. The title should be corrected to "Crystal structure of Cry78Aa, a novel protein from *Bacillus thuringiensis*, implies its new insecticidal activity on rice planthopper."
2. Line 103: "Therefore, Cry78Aa is likely to be a Bt protein with a novel structure or insecticidal mechanism." This statement is rather overstated as the similar structures are found in other Bt proteins e.g Cry35Ab and parasporin-2.
3. The isolated pore-forming domain Cry78Aa155-359 has been proposed as a pore-forming domain based on the structural homology with that of aerolysin-type toxin family. To confirm this hypothesis, this domain should be tested the pore-forming activity either in vitro or in vivo as has been demonstrated in other aerolysin-like toxins.
4. Lines 233-235: monomeric Cry78Aa155-359 can easily transform into oligomer or aggregate in 0.1% SDS condition, with around 30% of the total sample loaded onto each lane. These results suggest Cry78Aa is prone to oligomerization in a suitable condition in vivo. This may not be true as the condition in the sample buffer loaded in SDS-gel is totally different from that in the insect gut membrane environment. It is better to incubate Cry78Aa with the membrane extract of susceptible insect to observe the oligomeric complex.
5. Single mutations of residues proposed to be sugar-binding sites on Cry78Aa trefoil domain were found to reduce the mortality rate but at variable extents. The effects of these mutations on protein expression and conformation should be mentioned as well. Also, multiple mutations may be included to strongly confirm the role of these residues on sugar binding. Moreover, some other biophysical techniques may be used to confirm this sugar binding activity e.g surface plasmon resonance (SPR). SPR can also be used to confirm the binding of galactose or GalNAc with the trefoil domain of Cry78Aa compared with other sugar.
6. The putative MSR of Cry78Aa has been predicted based on the structural alignment. The presence of alternating hydrophobic and hydrophilic amino acid residues characteristic of MSR should also be demonstrated.
7. It should be demonstrated whether or not other common characteristics of aerolysin-type pore-forming toxins such as the presence of serine/threonine patches are present in the structure of Cry78Aa.
8. The language throughout this manuscript needs some work to edit, especially grammar mistakes, by someone with strong scientific English skills.

Reviewer #2 (Remarks to the Author):

The manuscript of Cao et al describes the structural characterization of Cry78Aa, a protein from *Bacillus thuringiensis* with insecticidal activity on rice planthopper. The structure of Cry78Aa resembles those of other toxins such as BinB and Cry235Ab1. It is composed of 2 domains, a carbohydrate recognition domain and a pore forming domain. The authors have convincingly shown (structure alignment and mutant generation) that the activity of Cry78Aa depends on the presence of galactose.

The work is sound and technically up-to-date. The experimental part is clearly described and the structural results are clearly presented. Unfortunately, no data is provided to explain the mechanism of action of Cry78Aa. It is shown that it can form high molecular weight oligomers. However, the structure of the final pore has not been investigated. In conclusion, this is a well conducted study but the results are limited in scope.

Response letter to editor and reviewers

COMMSBIO-21-1677-T

All responses are highlighted in green

Reviewer #1:

This reviewer thinks our manuscript suggest a mechanism underlying the function of Cry78Aa against rice planthopper and provide useful guidelines for further engineering of protein for maximizing the toxicity. At the same time, the reviewer also raised some concerns and gave us suggestions to improve our manuscript.

Comments:

- 1) *The title should be corrected to “Crystal structure of Cry78Aa, a novel protein from Bacillus thuringiensis, implies its new insecticidal activity on rice planthopper.”*

We thank the reviewer for pointing out this mistake. The title has been changed to “*Crystal structure of Cry78Aa, a novel protein from Bacillus thuringiensis, implies a new insecticidal activity on rice planthopper*”. We also feel sorry for other grammatical errors in the main text. We have corrected them all in the revised manuscript with the help of professionals in English scientific writing.

- 2) *Line 103: “Therefore, Cry78Aa is likely to be a Bt protein with a novel structure or insecticidal mechanism.” This statement is rather overstated as the similar structures are found in other Bt proteins e.g Cry35Ab and parasporin-2.*

We thank the reviewer for this comment. We state that Cry78Aa is likely to be a Bt protein with a novel insecticidal mechanism based on the following findings.

Firstly, Cry78Aa consists of the S-type ricin B lectin domain and the Toxin_10 domain. Although Cry78Aa and Cry35Ab have similar overall characteristics, there are still significant differences. The N-terminal domain of

Cry35Ab is a β -trefoil fold domain containing two repeated QxW motifs. The C-terminal domain of Cry35Ab contains six helices and three antiparallel β -sheets and ends in a C-terminal cluster of three α -helices. Meanwhile, we also know that Cry35Ab and Cry34Ab proteins are required to act together to produce toxicity to the western corn rootworm (WCR) *Diabrotica virgifera virgifera* via a pore-forming mechanism of action. In contrast, in our study, the Cry78Aa protein alone has high insecticidal activity against rice planthopper.

Besides, Cry78A is significantly different from parasporin-2 in structure. Parasporin-2 protein is a molecule that can be divided into three domains. Domain 1, which comprises a small β -sheet sandwiched by short α -helices, is probably the target-binding module. The other two domains are both β -sandwiches and are thought to be involved in oligomerization and pore formation. Domain 2 has a putative channel-forming β -hairpin characteristic of aerolysin-type toxins.

We fully understand the concern of the reviewer and agree with the point that this kind of conclusion should not be overstated. Thus, we have modified the sentence “*Therefore, Cry78Aa is likely to be a Bt protein with a novel structure or insecticidal mechanism.*” into “*Therefore, we sought to resolve the structure of Cry78Aa to determine whether it adopts a novel structure compared to other Bt toxin proteins.*”. in revised main text.

3) *The isolated pore-forming domain Cry78Aa₁₅₅₋₃₅₉ has been proposed as a pore-forming domain based on the structural homology with that of aerolysin-type toxin family. To confirm this hypothesis, this domain should be tested the pore-forming activity either in vitro or in vivo as has been demonstrated in other aerolysin-like toxins.*

We thank the reviewer for this instructive suggestion. Based on the sequence and structure comparison between Cry78Aa and other aerolysin family toxic proteins, we proposed that the C-terminal domain of Cry78Aa, Cry78Aa₁₅₅₋₃₅₉, is a pore-forming domain. The reviewer might have not

noticed that the CTD of Cry78Aa alone has no insecticidal activity towards target insect *L. striatellus* (**Figure 2c** in the main text), which mean it probably has no pore-forming activity by itself. The mixture of the purified NTD and CTD did not show any insecticidal activity as well. Only the intact Cry78Aa has insecticidal activity towards *L. striatellus*.

We understand it is very important to confirm that Cry78Aa indeed can form a pore structure and its pore-forming activity need to be tested. We have made many attempts *in vitro* and *in vivo* using the assays which have been demonstrated for other aerolysin-like toxins, but so far, the results were not satisfactory.

We also checked the conditions reported for the other aerolysin family toxic proteins to oligomerize *in vitro*. Specifically, we tried to induce Cry78Aa oligomerization by detergent, lipid and BBMV extracted from target insect larva (*L. striatellus*) *in vitro*, and then analyzed the different states of Cry78Aa in the process of pore formation by Cryo-EM. However, thus far, we haven't found any suitable conditions to make Cry78Aa enter a relatively stable oligomeric state or form a pore structure.

We also tried to construct the oligomeric state or pore structure of Cry78Aa by liposome. We successfully prepared the POPC liposome and found that Cry78Aa can be constructed on this kind of liposomes, for considerable proteins was extracted from liposome and detected be SDS-PAGE (Figure 1A). However, unfortunately, we could not observe evident oligomeric Cry78Aa or any pore structure (Figure 1B).

Figure 1. Construction of Cry78Aa oligomer or pore structure on POPC liposome

A. SDS-PAGE detection of oligomeric Cry78Aa on POPC liposome. After incubating Cry78Aa with 0.05% SDS for 16 h, mixed Cry78Aa with prepared POPC liposome at room temperature for 1 hour, then performed low-speed centrifugation at 6000 rpm for 10 min to remove aggregates, sample the precipitate P1 and supernatant S1; then ultracentrifuge sample S1 at 120,000 g for 2.5 h, sample the precipitate P2 and supernatant S2. Cry78Aa without POPC was set as a control. We can see oligomeric Cry78Aa was absent in sample S2 of Cry78Aa plus liposome (red arrow on the right), but exclusively showed in sample P2. At any time, sample of Cry78Aa only incubated with SDS partially existed as oligomeric state (red arrow on the left). **B.** Cryo-EM observation of Cry78Aa on liposome. Sample P2 was resuspended and observed by Cryo-EM. The morphology of Cry78Aa oligomers could not be clearly detected, red arrows in the red circle indicated the potential particles of Cry78Aa.

A previous study has report that the pore-forming proteins can form pore structures on liposome and be observed by Cryo-EM (Ruan et.al, 2018, Nature, Vol 557, page 62-67).

Figure 5. Visualization of GSDMD^{Nterm} pores in liposomes by cryo-electron microscopy.

A–C Cryo-electron micrographs of GSDMD^{Nterm} pores in *E. coli* polar lipid liposomes. The micrographs were acquired at protein/lipid molar ratios of 1/1,000, 1/500, and 1/100, respectively. Black arrows indicate ring-shaped structures corresponding to oligomeric GSDMD^{Nterm} pore forms. Scale bars = 80 nm.

D Proteoliposome with protein/lipid molar ratio of 1/100 at higher magnification. Black arrows indicate ring-shaped structures corresponding to oligomeric GSDMD^{Nterm} pore forms. Scale bar = 80 nm.

We failed to detect this phenomenon on the POPC liposome incubated with Cry78Aa. We prepared liposomes by mixing lipids derived from *E.coli* or soybean and incubated them with Cry78Aa respectively. Oligomeric Cry78Aa

or pore structure were not observed. We also prepared liposomes which enclosed fluorescent molecule calcein and incubated it with Cry78Aa, resembling the method used by other Bt toxins, to detect the membrane leakage resulted from pore structure formation of Cry78Aa. We have not got any ideal results thus far but we will keep optimizing the experimental procedures. We will try to extract the Cry78Aa oligomer from the liposome, and then take the sample to observe the morphology of the Cry78Aa oligomer directly under the Cryo-EM. Meanwhile, we keep in mind that the formation of the pore structure of Cry78Aa may need the help of suitable lipids or receptors which need to be further identified.

To investigate whether the pore structure formed by Cry78Aa on the cell surface could be observed, we fed the planthopper larvae with Cry78Aa protein and checked the intestinal tissue by section and transmission electron microscope. We found that after feeding Cry78Aa protein for 48 h, the midgut epithelial cells of *L. striatellus* were damaged to some extent, including digestion of microvilli, vacuolation of cytoplasm, swelling and fracture of endoplasmic reticulum and condensation of chromatin (Figure 2). However, both the control and the treatment groups showed holes at the base of midgut epithelium (blue box), and almost all the intestinal lumens contained unknown small particles (yellow box). It was difficult to observe relatively complete midgut epithelial cells and perimicrovilli matrix unique to hemipteran pests. We hypothesized that the midgut tissue was severely damaged due to improper dissection. We are currently trying to optimize the anatomy and sample preparation methods and will continue to analyze the damage of midgut caused by Cry78Aa protein.

Figure 2. The pathological changes after *L. striatellus* feeding on artificial diet containing 15 µg/mL Cry78Aa protein with 48 h.

We made every efforts and tried a lot of methods to test the pore-forming activity of Cry78Aa but they all failed at the current state. We speculate that it might be easier for Cry78Aa to transform into the pore structure with the help of the receptor, glycolipids or glycoproteins, on the surface of the cell membrane from the target insect. Thus, we attempted to identify the receptor of Cry78Aa and hope it facilitates the pore-forming activity assay, pore structure acquirement and insecticidal mechanism.

For example, we tried to analyze the Cry78Aa binding proteins from *Laodelphax striatellus* BBMV through the ligand fishing technology established in our laboratory (referring to the method of Zhou *et al.*). Similar results were obtained in two independent replicates (Table 1). The mass spectrometry assay results showed that many potential Cry78Aa binding proteins were detected, but common Bt protein receptors with high matching degrees were not detected. These potential binding proteins of Cry78Aa protein include some cytoskeletal proteins or membrane proteins, such as

tubulin, myosin, clathrin, etc. The specific functions of these potential receptor proteins still need to be further verified.

Table 1. The mass spectrometry assay results of Cry78Aa protein binding protein on BBMV of *Laodelphax striatellus* (LC-MS)

No.	Description	Mass (Da)	Score	Uniprot	Function
1	Uncharacterized protein	9118	64	A0A482X6H2	Cytochrome C oxidase subunit NDUFA4 subtype X3
2	Cytochrome b561 domain-containing protein	30800	55	A0A482XMU4	
3	Uncharacterized protein	23019	83	A0A482X1W6	Ras-associated protein RAB-10
4	FA_desaturase domain-containing protein	42791	71	A0A482XKD3	
5	Uncharacterized protein	215296	267*	A0A482XM11	Myosin
6	Clathrin heavy chain	193129	162*	A0A482X8P8	
7	Uncharacterized protein	92559	283*	A0A482X042	Adhesive plaque matrix protein
8	Tropomyosin 1	32869	321*	M9VSN8	
9	H(+)-transporting two-sector	66366	242*	A0A482X357	
10	NADH dehydrogenase	32484	57	A0A482XE63	
11	Alpha 1-tubulin	50570	148*	Q6R7Z0	
12	Tubulin alpha chain	40024	94	A0A482X0K8	
13	Prohibitin	35875	72	A0A482XDH2	
14	Elongation factor 1-alpha	50814	237*	A0A482WR01	
15	Vacuolar proton pump subunit B	55787	135*	A0A482X845	
16	Uncharacterized protein	34693	90	A0A482XHD4	Retinol dehydrogenase 13-like
17	Uncharacterized protein	35191	68	A0A482WVK7	Putative tricarboxylic acid transporter, mitochondria
18	PHB domain-containing protein	48375	139*	A0A482XPC3	
19	Cytochrome P450	58778	76	A0A386RV61	
20	Uncharacterized protein	52209	52	A0A482X5H7	Troponin
21	Transket_pyr domain-containing protein	68811	64	A0A482XNB7	
22	Uncharacterized protein	40791	54	A0A482X6S7	Guanine nucleotide binding protein G(O) subunit alpha
23	Uncharacterized protein	32551	72	A0A482WVG3	
24	Uncharacterized protein	50273	71	A0A482WIL8	Acetoacetyl CoA thiolase
25	ATP synthase subunit d, mitochondrial	18960	30	A0A482XEE2	
26	Uncharacterized protein	16699	50	A0A482WQD9	
27	Uncharacterized protein	6376	40	A0A482WRW6	Mitochondrial proton transport ATP synthase complex
28	Protein quiver	19411	34	A0A482WZG1	
29	Uncharacterized protein	20589	60	A0A482X9W4	
30	Complex I-B17	18122	43	A0A482XG22	Mitochondrial respiratory chain complex I
31	Small heat shock protein 4	20500	54	A0A1S5VZD0	
32	Uncharacterized protein	22752	73	A0A482X2W8	
33	Glutathione S-transferase S3	23312	51	H2DMI0	

Note: Scores marked with asterisk were significant.

We found that Cry78Aa bind to midgut BBMV of *L. striatellus*, and stable oligomers could form during the binding process. At the same time, the binding of Cry78Aa protein to BBMV showed a significant dose effect, but the binding did not reach the saturation trend even though the concentration of Cry78Aa protein was up to 2700 nmol/L (Figure 3). The number of membrane protein receptors in insect midgut cells is limited and their binding to Cry proteins is usually rapid to saturation. Therefore, we speculate that Cry78Aa may interact with abundant membrane lipids on the cell membrane.

Figure 3. The binding analysis results of Cry78Aa protein and BBMV of *L. striatellus*. The red arrow represents the Cry78Aa oligomer. Lanes 1-7 represent 0, 11.1, 33.3, 100, 300, 900, 2700 nmol/L Cry78Aa protein binding to BBMV, respectively. Lane 8: negative control Cry78Aa protein only. CK+: positive control 5 ng Cry78Aa protein.

We extracted lipids from midgut of target insects (*Laodelphax striatellus*), non-target insects (*Spodoptera frugiperda* and *Holotrichia parallela*), and the unknown target of Hemiptera (*Apolygus lucorum*), and analyzed their lipid composition by UHPLC-ESI-MS/MS. The results showed that 38 lipid subgroups and 2573 lipid molecules were identified in midgut lipid extracts of the four insects. However, the relative abundance of lipids in midgut of *L. striatellus* was significantly different from that of the other three insects (Figure 4). Compared with the other three species, the relative abundance of phosphatidylethanolamine (PE), ceramide (Cer), sphingosine (SPH) and diacylglycerol (DG) in the midgut of *L. striatellus* was significantly higher (red arrow), while the abundance of triglyceride (TG) was significantly lower (blue arrow). It is noteworthy that the 1-hydroxyl group of ceramides is glycosylated

to form glycosphingolipid, while the latter is the monosaccharide molecules including D-galactose, N-acetyl galactosamine, D-glucose, fucose and sialic acid and so on. Although their abundance is relatively low, several glycosphingolipids were detected in the midgut lipids of *L. striatellus*, including CerG2GNAc1, ganglioside (GD3, GM1, GM3), monohexosylceramide (Hex1Cer), dihexosylceramide (Hex2Cer), trihexosylceramide (Hex3Cer), sulfatide (ST). Among them, the relative abundance or molecular composition of CerG2GNAc1, GM3 and hexose ceramide were different from the other three pests. The glycosphingolipid has been reported as Cry5B protein receptors in nematode, and Bt toxin can identify conservative sites "four sugar core". In addition, gangliosides have also been reported as receptors for cholera toxin, tetanus toxin, galactose lectin 1, etc. Currently, we are trying to isolate these lipids and analyze their interaction with Cry78Aa.

Figure 4. Results of lipid composition analysis in midgut of four insects. The gray, green, black and yellow lines represent *Laodelphax striatellus* (Ls), *Apolygus lucorum* (Al), *Holotrichia parallela* (Hp) and *Spodoptera frugiperda* (Sf), respectively.

4) Lines 233-235: monomeric Cry78Aa₁₅₅₋₃₅₉ can easily transform into oligomer or aggregate in 0.1% SDS condition, with around 30% of the total sample loaded onto each lane. These results suggest Cry78Aa is prone to oligomerization in a suitable condition in vivo. This may not be true as the condition in the sample buffer loaded in SDS-gel is totally different from that in the insect gut membrane environment. It is better to incubate

Cry78Aa with the membrane extract of susceptible insect to observe the oligomeric complex.

We thank the reviewer for this instructive suggestion. The full-length Cry78Aa was incubated with extracts of the midgut membrane of *Laodelphax striatellus*, and the oligomer status was detected by electrophoresis and western blot. The results showed that co-incubation with midgut membrane extract of *L.striatellus* did make Cry78Aa form a larger oligomer complex. In contrast, the oligomer was not detected in sample of Cry78Aa or BBMV alone using the same method. The results suggested that appropriate physiological condition is probably needed for monomeric Cry78Aa to transform into oligomer in. These results have been added as **Figure 2e** and described by a new paragraph *“To analyze the oligomerization status of Cry78Aa protein under physiological conditions, the oligomer formation of Cry78Aa was studied by incubating Cry78Aa protein with brush border membrane vesicles (BBMVs) from the sensitive pest Laodelphax striatellus.....”* in the revised manuscript. We believe this finding significantly enhanced our conclusion.

We also modified the sentence *“These results suggest Cry78Aa is prone to oligomerization in a suitable condition in vivo”* into *“These results suggested that Cry78Aa is prone to oligomerization under suitable conditions.”*.

5) *Single mutations of residues proposed to be sugar-binding sites on Cry78Aa trefoil domain were found to reduce the mortality rate but at variable extents. The effects of these mutations on protein expression and conformation should be mentioned as well. Also, multiple mutations may be included to strongly confirm the role of these residues on sugar binding. Moreover, some other biophysical techniques may be used to confirm this sugar binding activity e.g surface plasmon resonance (SPR). SPR can also be used to confirm the binding of galactose or GalNAc with the trefoil domain of Cry78Aa compared with other sugar.*

We sincerely thank the reviewer for the kind reminders. Theoretically, the effect of mutation on protein folding of Cry78Aa could exist and the possibility that the decreased insecticidal activity of mutated Cry78Aa toward target insect larva resulted from destroyed conformation rather than damaged interaction between protein and sugar should be excluded. During the expression and purification of wild and mutated Cry78Aa, all the proteins have good yield and behavior in gel filtration, which suggested they were well folded (Figure 5). In order to avoid misunderstanding, we have added the sentence *“During the expression and purification procedures, all the mutated Cry78Aa proteins exhibited good yield and behavior, implying that they were well folded.”* in the revised manuscript. Unfortunately, it is not convenient for us to perform new bioassay experiments owing to the COVID-19 epidemic situation. We apologize for it but currently we cannot show the role of multiple mutations on sugar binding. We believe that single-point mutants have proved that these amino acids are the key sites for Cry78Aa to exert insecticidal activity.

Figure 5. Gel filtration profiles of wild Cry78Aa and the mutants during purification.

In addition, although the binding affinity is relatively low, we have confirmed by the isothermal titration calorimetry (ITC) method that Cry78Aa can specifically bind to carbohydrates containing galactose bonds, including lactose, galactose and GalNAc. Meanwhile, the binding of Cry78Aa to other carbohydrates such as glucose, GlcNAc, arabinose and mannose is undetectable. These results are completely consistent with the results of the sugar protection bioassay, revealing the role of specific types of glycoproteins or glycolipids in the recognition of membrane surface signals by Cry78Aa. We have added the results in the revised manuscript as **supplementary Figure 7**. We also added some sentences “*Isothermal titration calorimetry (ITC) experiments confirmed that Cry78Aa indeed bound to galactose, GalNAc and lactose, and the fitted binding dissociation constants (K_d) were approximately 15.5, 9.8 and 10.2 mM, respectively (Supplementary Fig. 7a-c). Meanwhile, the other carbohydrates, such as glucose, GlcNAc, mannose and arabinose, had no detectable interaction with Cry78Aa (Supplementary Fig. 7d-g)*” to describe the results. We believe these results strongly enhanced our conclusion.

6) *The putative MSR of Cry78Aa has been predicted based on the structural alignment. The presence of alternating hydrophobic and hydrophilic amino acid residues characteristic of MSR should also be demonstrated.*

We sincerely thank the reviewer for the kind reminders. According to amino acid sequence analysis, it was confirmed that the putative MSR of Cry78Aa had the characteristics of alternating hydrophobic and hydrophilic residues. We demonstrated this feature of Cry78Aa in added diagrams (**Figure 6a, b**) and described it in the revised main text, by adding two sentences “*The putative MSR of Cry78Aa consisted of alternating hydrophobic and hydrophilic amino acid residues (Fig. 6a, b). This feature attributes the MSR of Cry78Aa an amphiphilic property which is commonly shared among aerolysin family toxins*”.

7) *It should be demonstrated whether or not other common characteristics of aerolysin-type pore-forming toxins such as the presence of serine/threonine patches are present in the structure of Cry78Aa.*

We sincerely thank the reviewer for the kind reminders. We indeed found the existence of the serine/threonine patches in the CTD of Cry78Aa. We have drawn some new diagrams to demonstrate this feature and added the analysis as **Figure 6c and 6d**. We also added some sentences “*There also existed serine/threonine patches in the CTD of Cry78Aa (Fig. 6c), similar to aerolysin-type pore-forming toxins*” in the revised main text. We believed this analysis could help the readers better understand the structural features of MSR or CTD of Cry78Aa.

8) *The language throughout this manuscript needs some work to edit, especially grammar mistakes, by someone with strong scientific English skills.*

Thanks again for this kindly advice. We have checked all the grammar errors in the main text and corrected them in the revised manuscript with the help of professionals in scientific writing.

Reviewer #2:

This reviewer thinks our manuscript has convincingly shown (structure alignment and mutant generation) that the activity of Cry78Aa depends on the presence of galactose. The work is sound and technically up-to-date. The experimental part is clearly described and the structural results are clearly presented. At the same time, the reviewer also raised some weaknesses that the manuscript needs to be further improved.

Comments:

1. Unfortunately, no data is provided to explain the mechanism of action of Cry78Aa. It is shown that it can form high molecular weight oligomers. However, the structure of the final pore has not been investigated. In conclusion, this is a well conducted study but the results are limited in scope.

We thank the reviewer for his/her recognition of our experimental work and display results in this manuscript. At the same time, the reviewer believes that this paper lacks the display of relevant data on the research of Cry78Aa insecticidal mechanism. Therefore, at this stage, "the results are limited in scope". In fact, Cry78Aa's structural analysis and insecticidal mechanism research were always simultaneously performed in our study. The purpose of resolving the protein structure is also to lay a foundation for the study of insecticidal mechanism and subsequent design and application of insecticidal protein. We have made every efforts to clarify the mechanism underlying the killing of rice planthopper by Cry78Aa, as well as to resolve the final pore structure of Cry78Aa. However, the progress is not satisfactory and could not be presented in main text at present stage.

We checked the conditions reported for the other aerolysin family toxic proteins to oligomerize in vitro.. Specifically, we tried to induce Cry78Aa oligomerization by detergent, lipid and BBMV extracted from target insect larva (*L. striatellus*) in vitro, and then analyzed the different states of Cry78Aa

in the process of pore formation by Cryo-EM. However, thus far, we haven't found any suitable conditions to make Cry78Aa enter a relatively stable oligomeric state or form a pore structure.

We also tried to construct the oligomeric state or pore structure of Cry78Aa by liposome. We successfully prepared the POPC liposome and found that Cry78Aa can be constructed on this kind of liposomes, for considerable proteins was extracted from liposome and detected by SDS-PAGE (Figure 1A). However, unfortunately, we could not observe evident oligomeric Cry78Aa or pore structure (Figure 1B).

Figure 1. Construction of Cry78Aa oligomer or pore structure on POPC liposome

A. SDS-PAGE detection of oligomeric Cry78Aa on POPC liposome. After incubating Cry78Aa with 0.05% SDS for 16 h, mixed Cry78Aa with prepared POPC liposome at room temperature for 1 hour, then performed low-speed centrifugation at 6000 rpm for 10 min to remove aggregates, sample the precipitate P1 and supernatant S1; then ultracentrifuge sample S1 at 120,000 g for 2.5 h, sample the precipitate P2 and supernatant S2. Cry78Aa without POPC was set as a control. We can see oligomeric Cry78Aa was absent in sample S2 of Cry78Aa plus liposome (red arrow on the right), but exclusively showed in sample P2. At any time, sample of Cry78Aa only incubated with SDS partially existed as oligomeric state (red arrow on the left). **B.** Cryo-EM observation of Cry78Aa on liposome. Sample P2 was resuspended and observed by Cryo-EM. The morphology of Cry78Aa oligomers could not be clearly detected, red arrows in the red circle indicated the potential particles of Cry78Aa.

A previous study has reported that the pore-forming proteins can form pore structures on liposomes and be observed by Cryo-EM (Ruan et al., 2018, *Nature*, Vol 557, page 62-67).

Figure 5. Visualization of GSDMD^{Nterm} pores in liposomes by cryo-electron microscopy.

A–C Cryo-electron micrographs of GSDMD^{Nterm} pores in *E. coli* polar lipid liposomes. The micrographs were acquired at protein/lipid molar ratios of 1/1,000, 1/500, and 1/100, respectively. Black arrows indicate ring-shaped structures corresponding to oligomeric GSDMD^{Nterm} pore forms. Scale bars = 80 nm.
 D Proteoliposome with protein/lipid molar ratio of 1/100 at higher magnification. Black arrows indicate ring-shaped structures corresponding to oligomeric GSDMD^{Nterm} pore forms. Scale bar = 80 nm.

We failed to detect this phenomenon on the POPC liposome incubated with Cry78Aa. We prepared liposomes by mixing lipids derived from *E. coli* or soybean and incubated them with Cry78Aa respectively. Oligomeric Cry78Aa or pore structure were not observed. We also prepared liposomes which enclosed fluorescent molecule calcein and incubated it with Cry78Aa, resembling the method used by other Bt toxins, to detect the membrane leakage resulted from pore structure formation of Cry78Aa. We have not got any ideal results thus far but we will keep optimizing the experimental procedures. We will try to extract the Cry78Aa oligomer from the liposome, and then take the sample to observe the morphology of the Cry78Aa oligomer directly under the Cryo-EM. Meanwhile, we keep in mind that formation of the pore structure of Cry78Aa may need the help of suitable lipids or receptors which need to be further identified.

To investigate whether the pore structure formed by Cry78Aa on the cell surface could be observed, we fed the planthopper larvae with Cry78Aa protein and checked the intestinal tissue by section and transmission electron microscope. We found that after feeding Cry78Aa protein for 48 h, the midgut epithelial cells of *L. striatellus* were damaged to some extent, including digestion of microvilli, vacuolation of cytoplasm, swelling and fracture of endoplasmic reticulum and condensation of chromatin (Figure 2). However, both the control and the treatment groups showed holes at the base of midgut epithelium (blue box), and almost all the intestinal lumens contained unknown

small particles (yellow box). It was difficult to observe relatively complete midgut epithelial cells and perimicrovilli matrix unique to hemipteran pests. We hypothesized that the midgut tissue was severely damaged due to improper dissection. We are currently trying to optimize the anatomy and sample preparation methods and will continue to analyze the damage of midgut caused by Cry78Aa protein.

Figure 2. The pathological changes after *L. striatellus* feeding on artificial diet containing 15 μg/mL Cry78Aa protein with 48 h.

We speculate that it might be easier for Cry78Aa to transform into the pore structure with the help of the receptor, glycolipids or glycoproteins, on the surface of the cell membrane from the target insect. Thus, we attempted to identify the receptor of Cry78Aa and hope it facilitates the pore-forming activity assay, pore structure acquisition and insecticidal mechanism.

For example, we tried to analyze the Cry78Aa binding proteins from *Laodelphax striatellus* BMV the ligand fishing technology established in our laboratory (referring to the method of Zhou *et al.*). Similar results were obtained in two independent replicates (Table 1). The mass spectrometry

assay results showed that many potential Cry78Aa binding proteins were detected, but common Bt protein receptors with high matching degree were not detected. These potential binding proteins of Cry78Aa protein include some cytoskeletal proteins or membrane proteins, such as tubulin, myosin, clathrin, etc. The specific functions of these potential receptor proteins still need to be further verified.

Table 1. The mass spectrometry assay results of Cry78Aa protein binding protein on BBMV of *Laodelphax striatellus* (LC-MS)

No.	Description	Mass (Da)	Score	Uniprot	Function
1	Uncharacterized protein	9118	64	A0A482X6H2	Cytochrome C oxidase subunit NDUFA4 subtype X3
2	Cytochrome b561 domain-containing protein	30800	55	A0A482XMU4	
3	Uncharacterized protein	23019	83	A0A482X1W6	Ras-associated protein RAB-10
4	FA_desaturase domain-containing protein	42791	71	A0A482XKD3	
5	Uncharacterized protein	215296	267*	A0A482XM11	Myosin
6	Clathrin heavy chain	193129	162*	A0A482X8P8	
7	Uncharacterized protein	92559	283*	A0A482X042	Adhesive plaque matrix protein
8	Tropomyosin 1	32869	321*	M9VSN8	
9	H(+)-transporting two-sector	66366	242*	A0A482X357	
10	NADH dehydrogenase	32484	57	A0A482XE63	
11	Alpha 1-tubulin	50570	148*	Q6R7Z0	
12	Tubulin alpha chain	40024	94	A0A482X0K8	
13	Prohibitin	35875	72	A0A482XDH2	
14	Elongation factor 1-alpha	50814	237*	A0A482WR01	
15	Vacuolar proton pump subunit B	55787	135*	A0A482X845	
16	Uncharacterized protein	34693	90	A0A482XHD4	Retinol dehydrogenase 13-like
17	Uncharacterized protein	35191	68	A0A482WVK7	Putative tricarboxylic acid transporter, mitochondria
18	PHB domain-containing protein	48375	139*	A0A482XPC3	
19	Cytochrome P450	58778	76	A0A386RV61	
20	Uncharacterized protein	52209	52	A0A482X5H7	Troponin
21	Transket_pyr domain-containing protein	68811	64	A0A482XNB7	
22	Uncharacterized protein	40791	54	A0A482X6S7	Guanine nucleotide binding protein G(O) subunit alpha
23	Uncharacterized protein	32551	72	A0A482WVG3	
24	Uncharacterized protein	50273	71	A0A482WIL8	Acetoacetyl CoA thiolase
25	ATP synthase subunit d, mitochondrial	18960	30	A0A482XEE2	
26	Uncharacterized protein	16699	50	A0A482WQD9	
27	Uncharacterized protein	6376	40	A0A482WRW6	Mitochondrial proton transport ATP synthase complex
28	Protein quiver	19411	34	A0A482WZG1	
29	Uncharacterized protein	20589	60	A0A482X9W4	
30	Complex I-B17	18122	43	A0A482XG22	Mitochondrial respiratory chain complex I

31	Small heat shock protein 4	20500	54	A0A1S5VZD0
32	Uncharacterized protein	22752	73	A0A482X2W8
33	Glutathione S-transferase S3	23312	51	H2DMI0

Note: Scores marked with asterisk were significant.

We found that Cry78Aa bind to midgut BBMV of *L. striatellus*, and stable oligomers could form during the binding process. At the same time, the binding of Cry78Aa protein to BBMV showed a significant dose effect, but the binding did not reach the saturation trend even though the concentration of Cry78Aa protein was up to 2700 nmol/L (Figure 3). The number of membrane protein receptors in insect midgut cells is limited and their binding to Cry proteins is usually rapid to saturation. Therefore, we speculate that Cry78Aa may interact with abundant membrane lipids on the cell membrane.

Figure 3. The binding analysis results of Cry78Aa protein and BBMV of *L. striatellus*. The red arrow represents the Cry78Aa oligomer. Lanes 1-7 represent 0, 11.1, 33.3, 100, 300, 900, 2700 nmol/L Cry78Aa protein binding to BBMV, respectively. Lane 8: negative control Cry78Aa protein only. CK+: positive control 5 ng Cry78Aa protein.

We extracted lipids from midgut of target insects (*Laodelphax striatellus*), non-target insects (*Spodoptera frugiperda* and *Holotrichia parallela*), and the unknown target of Hemiptera (*Apolygus lucorum*), and analyzed their lipid composition by UHPLC-ESI-MS/MS. The results showed that 38 lipid subgroups and 2573 lipid molecules were identified in midgut lipid extracts of the four insects. However, the relative abundance of lipids in midgut of *L. striatellus* was significantly different from that of the other three insects (Figure 4). Compared with the other three species, the relative abundance of phosphatidylethanolamine (PE), ceramide (Cer), sphingosine (SPH) and diacylglycerol (DG) in the midgut of *L. striatellus* was significantly higher (red

arrow), while the abundance of triglyceride (TG) was significantly lower (blue arrow). It is noteworthy that the 1-hydroxyl group of ceramides is glycosylated to form glycosphingolipid, while the latter is the monosaccharide molecules including D-galactose, N-acetyl galactosamine, D-glucose, fucose and sialic acid and so on. Although their abundance is relatively low, several glycosphingolipids were detected in the midgut lipids of *L. striatellus*, including CerG2GNAc1, ganglioside (GD3, GM1, GM3), monohexosylceramide (Hex1Cer), dihexosylceramide (Hex2Cer), trihexosylceramide (Hex3Cer), sulfatide (ST). Among them, the relative abundance or molecular composition of CerG2GNAc1, GM3 and hexose ceramide were different from the other three pests. The glycosphingolipid has been reported as Cry5B protein receptors in nematode, and Bt toxin can identify conservative sites "four sugar core". In addition, gangliosides have also been reported as receptors for cholera toxin, tetanus toxin, galactose lectin 1, etc. Currently, we are trying to isolate these lipids and analyze their interaction with Cry78Aa.

Figure 4. Results of lipid composition analysis in midgut of four insects. The gray, green, black and yellow lines represent *Laodelphax striatellus* (Ls), *Apolygus lucorum* (Al), *Holotrichia parallela* (Hp) and *Spodoptera frugiperda* (Sf), respectively.

In summary, we have tried many methods to obtain the oligomeric state of Cry78Aa and attempt to resolve its pore forming structure. At the same time, we have also explored the molecular mechanism of its selective killing of

Planthopper. At present, we can only get some preliminary results and they can not be fully described in this paper.

At this stage, in the revised manuscript we supplemented the experimental results of Cry78Aa oligomerization induced by BBMV in the midgut of gray planthopper *in vitro*, which proved that Cry78Aa also tends to play its role through oligomerization under appropriate physiological conditions. We also added some ITC experimental results to prove that Cry78Aa can bind galactose, N-acetyl-galactosamine, lactose and has no interaction with glucose, arabinose and mannose, etc. These results are completely consistent with the results of sugar protection bioassay, revealing the role of specific types of glycoproteins or glycolipids in the recognition of membrane surface signals by Cry78Aa.

In the revised manuscript, we also analyzed the sequence and structure of the C-terminal pore forming domain of Cry78Aa in detail, and found some characteristics that are conserved in the pore forming domain of aerolysin family toxic proteins. These analyses helps us better understand the mechanism underlying Cry78Aa's oligomerization, pore forming and killing of target pests, and provide a basis for designing Cry78Aa protein with higher toxicity.

Combined with the previous experimental results, our manuscript mainly reports the crystal structure of the new Bt insecticidal protein Cry78Aa. According to the structural information, the role of its N-terminal clover domain in identifying the sugar signal on the surface of intestinal membrane cells in the target pest was revealed. Its function was verified by bioassays. At the same time, the process of oligomerization was speculated according to the structural characteristics of the C-terminal pore forming domain. We believe that our work not only lays a solid foundation for clarifying the specific insecticidal mechanism of Cry78Aa, but also provides a basis for its artificial design and application. This study has broad implications for the research field of Cry toxin protein.

REVIEWERS' COMMENTS:

Reviewer #1 (Remarks to the Author):

Reviewer's comments: The revised manuscript COMMSBIO-21-1677A of Cao et al. has addressed most of major concerns for manuscript publication and included some new experimental data to improve the research quality and strengthen the insights into the mechanism underlying the insecticidal activity of Cry78Aa toxin. Overall, the revised manuscript is of high quality for publication and makes an important contribution to the field of Bt-based biological control.

Reviewer #3 (Remarks to the Author):

The authors responded to the various points raised by the reviews. Two experimental parts have been added, one concerning the formation of oligomers in the presence of insect gut membrane environment, and the other concerning the ITC study of the interaction between the N-terminal trefoil domain and several types of sugars.

The authors have demonstrated quite convincingly the presence of high molecular weight oligomers. Unfortunately no further characterization was undertaken. The results of the binding experiments are a bit surprising. Indeed, affinities of the order of ten millimolar can be considered as very weak. Similar studies on other trefoil domains show rather affinities in the order of tens micromolar, i.e. a thousand times higher. This is reflected in the huge quantities of sugars (galactose) used for the sugar protection assays (mass ratio of 2000 :1 so molar ratio more than 400000). Given this, I fully agree with the authors' remark in the discussion section : « There is a concern about whether binding of the trefoil domain of Cry78Aa to carbohydrates on the surface of the cell membrane is enough to trigger oligomerization and the subsequent drastic conformational change of each protomer, which drives the drilling process. » as well as the one made a little later: « Thus, receptors other than glycoproteins or glycolipids on the membrane could also be essential for the effects of Cry78Aa on rice planthoppers. » The identification of such a receptor would obviously have highly increased the interest of the article. The description of the mechanism is based entirely on studies done on other pore forming toxins and therefore remains purely theoretical. As the authors point out: « Much of the aerolysin protein family has been investigated in both monomeric and pore states ». It is regrettable that this is not the case in the present study.

In conclusion, one may wonder what the present study really adds about Cry78Aa compared to the paper by Wang et al published in 2018. Of course the experimental structure by X-ray crystallography and the molecular studies on the mutants provide a definitive confirmation on the monomer topology and on the role of the two domains that compose the toxin. But the lack of data on the pore state and on the existence of a specific receptor greatly reduce the scope of the study. To conclude I can only agree with the authors: « More exploration related to mechanism elucidation and protein engineering is needed to maximize the usage of this promising toxin in agriculture. »

Response letter to editor and reviewers

COMMSBIO-21-1677-A

All responses are highlighted in green

Reviewer #1:

This reviewer believe that our revised manuscript has addressed most of major concerns for manuscript publication, has high publishing quality, and has made an important contribution to the field of Bt biological control.

Comments:

The revised manuscript COMMSBIO-21-1677A of Cao et al. has addressed most of major concerns for manuscript publication and included some new experimental data to improve the research quality and strengthen the insights into the mechanism underlying the insecticidal activity of Cry78Aa toxin. Overall, the revised manuscript is of high quality for publication and makes an important contribution to the field of Bt-based biological control.

We are very grateful to the reviewers for their previous valuable comments and guidance on our manuscript, which further improved the quality of our manuscript.

Reviewer #3:

This reviewer thinks that our revised manuscript responds to the various points raised in the review and convincingly proves the existence of high molecular weight oligomers. He/She affirmed the results of using experimental structure of X-ray crystallography and molecular studies of mutants to explain the monomer topology and on the role of the two domains that compose the toxin. At the same time, this reviewer believe that our study lacks data on the pore state and whether on the existence of a specific receptor, and more exploration related to mechanism clarification and protein engineering is needed to maximize the use of this promising toxin in agriculture.

Comments:

- 1. The authors have demonstrated quite convincingly the presence of high molecular weight oligomers. Unfortunately no further characterization was undertaken. The results of the binding experiments are a bit surprising. Indeed, affinities of the order of ten millimolar can be considered as very weak. Similar studies on other trefoil domains show rather affinities in the order of tens micromolar, i.e. a thousand times higher. This is reflected in the huge quantities of sugars (galactose) used for the sugar protection assays (mass ratio of 2000 :1 so molar ratio more than 400000). Given this, I fully agree with the authors' remark in the discussion section: There is a concern about whether binding of the trefoil domain of Cry78Aa to carbohydrates on the surface of the cell membrane is enough to trigger oligomerization and the subsequent drastic conformational change of each protomer, which drives the drilling process. » as well as the one made a little later: Thus, receptors other than glycoproteins or glycolipids on the membrane could also be essential for the effects of Cry78Aa on rice planthoppers. The identification of such a receptor would obviously have highly increased the interest of the article.*

We quite agree with the reviewer on the results of the "sugar binding" experiment. So far, we have only found that Cry78Aa has weak interaction

with several specific carbohydrates. Therefore, based on its structure and mutant activity, a possibility of insecticidal mechanism has been proposed. Of course, we are actively seeking specific receptors for Cry78Aa protein and further elucidating the specific role that carbohydrates play in receptor recognition, which does not rule out the existence of other important receptors on the membrane of rice planthoppers.

2. *The description of the mechanism is based entirely on studies done on other pore forming toxins and therefore remains purely theoretical. As the authors point out: Much of the aerolysin protein family has been investigated in both monomeric and pore states. It is regrettable that this is not the case in the present study.*

We agree with the reviewer. Indeed, this paper focuses on the crystal structure of Cry78Aa, an important Bt protein that kills rice planthopper, and preliminarily explains the relationship between the two domains of Cry78Aa from the theoretical level. Limited by the biological characteristics of rice planthopper (small individuals, lack of full-fledged cell lines, etc.) and the pore formation structure of Bt toxin, the pore structure of Cry78Aa protein cannot be well solved at present.

3. *In conclusion, one may wonder what the present study really adds about Cry78Aa compared to the paper by Wang et al published in 2018. Of course the experimental structure by X-ray crystallography and the molecular studies on the mutants provide a definitive confirmation on the monomer topology and on the role of the two domains that compose the toxin. But the lack of data on the pore state and on the existence of a specific receptor greatly reduce the scope of the study. To conclude I can only agree with the authors: More exploration related to mechanism elucidation and protein engineering is needed to maximize the usage of this promising toxin in agriculture.*

Thank the reviewer for their recognition of our experimental work and the results presented in this paper. We couldn't agree more with the

reviewer's opinion that “*the lack of data on the pore state and on the existence of a specific receptor greatly reduce the scope of the study*” . Unfortunately, limited by the biological control of rice planthopper and the study of Bt toxin pore morphology, our results are very limited at this stage. And the crystal structure and mutants of Cry78Aa protein can only help to discover the potential insecticidal mechanism of Cry78Aa protein to a certain extent. We are actively investigating its pore state and specific receptor by any available methods, and there are some preliminary results, but more research is needed to better elucidate the mechanism of this protein to maximize the use of this promising toxin in agriculture. We thank the reviewer for his/her attention to our study and look forward to solving the pore structure or specifying specific receptors in the near future, when we will submit and share our research results again.